# Tailor3D: Customized 3D Assets Editing and Generation with Dual-Side Images

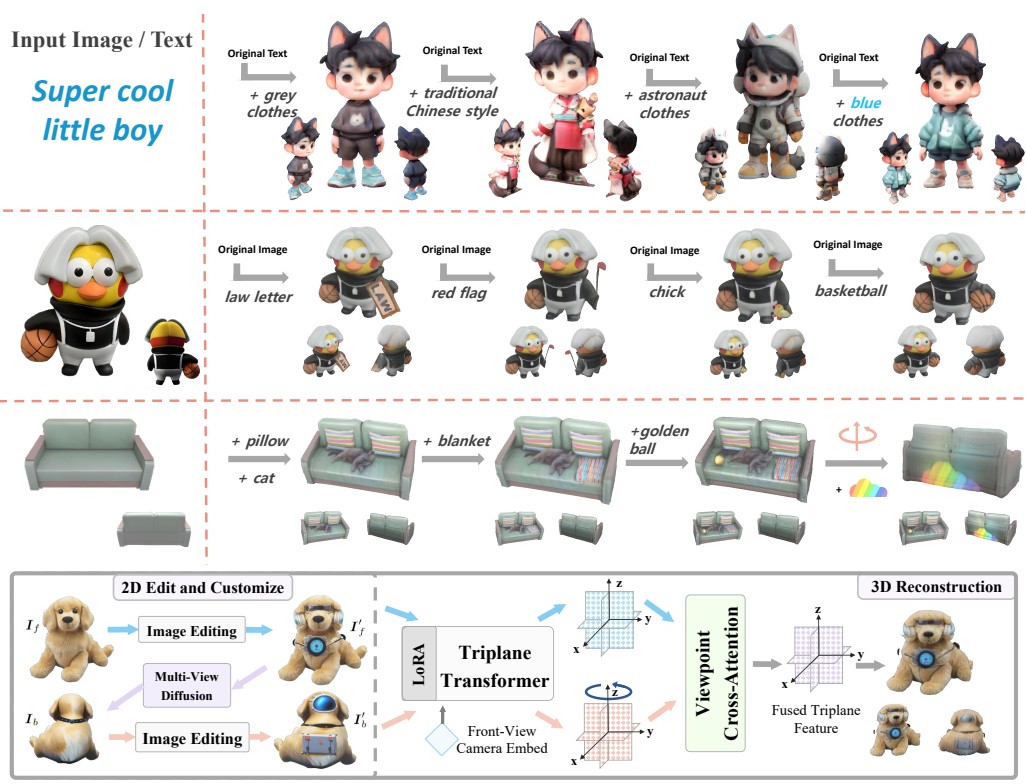

Figure 1: **Results and Pipeline**. We show our method for 3D style customization, as well as geometry and texture editing. Our pipeline involves editing images and generating the 3D object using Dual-sided LRM, with each step completed in just 5s, allowing for rapid 3D object customization.

## Abstract

Recent advances in 3D AIGC have shown promise in directly creating 3D objects from text and images, offering significant cost savings in animation and product design. However, detailed edit and customization of 3D assets remains a long-standing challenge. Specifically, 3D Generation methods lack the ability to follow finely detailed instructions as precisely as their 2D image creation counterparts. Imagine you can get a toy through 3D AIGC but with undesired accessories and dressing. To tackle this challenge, we propose a novel pipeline called Tailor3D, which swiftly creates customized 3D assets from editable dual-side images. We aim to emulate a tailor's ability to locally change objects or perform overall style transfer. Unlike creating 3D assets from multiple views, using dual-side images eliminates conflicts on overlapping areas that occur when editing individual views. Specifically, it begins by editing the front view, then generates the back view of the object through multi-view diffusion. Afterward, it proceeds to edit the back views. Finally, a Dual-sided LRM is proposed to seamlessly stitch together the front and back 3D features, akin to a tailor sewing together the front and back of a garment. The Dual-sided LRM rectifies imperfect consistencies between the front and back views, enhancing editing capabilities and reducing memory burdens while seamlessly integrating them into a unified 3D representation with the

LoRA Triplane Transformer. Experimental results demonstrate Tailor3D's effectiveness across various 3D generation and editing tasks, including 3D generative fill and style transfer. It provides a user-friendly, efficient solution for editing 3D assets, with each editing step taking only seconds to complete.

# 1 INTRODUCTION

In recent years, technologies like Stable Diffusion Rombach et al. (2022) and ControlNet Zhang et al. (2023) have revolutionized 2D AI-generated content (AIGC), making tasks like text-to-image synthesis, image editing, and style transfer more accessible and efficient. Concurrently, the potential of 3D AIGC has been recognized, allowing for the direct generation of 3D objects by integrating text and images, significantly reducing costs. Early optimization-based methods Xu et al. (2023); Poole et al. (2023); Wang et al. (2023b), where each object needs to be individually optimized, used multi-view stable diffusion Liu et al. (2024b;c); Sun et al. (2024) which means generating images of an object from multiple perspectives by inputting an image from one perspective—to produce fine-grained objects but were slow, taking minutes to hours. However, feed-forward methods leveraging large-scale 3D asset datasets Deitke et al. (2023) and Transformer models now enable the creation of high-quality 3D objects in seconds. Despite progress in generation, advancements in 3D customization and editing, such as adding patterns or changing styles of 3D objects, are still scarce.

In Feed-Forward Methods, although LRM Hong et al. (2024) can generate high-quality 3D objects from a single view, it often lacks comprehensive details from other perspectives. In contrast, techniques like Instant3D Li et al. (2024) and LGM Tang et al. (2024a) use multi-view diffusion Shi et al. (2023b); Wang & Shi (2023) to generate images from four perspectives (front, back, left, and right) before reconstruction. While increasing the number of perspectives can capture more visual information, it also brings some challenges: managing multiple views simultaneously increases the complexity of editing tasks. For instance, if we want to change the color of a specific part of the object, it is difficult to precisely correspond the changes across all four images. To balance the richness of visual information and the ease of editing, we recommend prioritizing the front and back views. These views typically contain comprehensive information about the object and have minimal overlap, allowing them to be edited independently, thus simplifying operations.

We propose an efficient and user-friendly 3D rapid editing framework, Tailor3D, which introduces a novel 3D editing way by leveraging advanced 2D image editing techniques. This framework delegates the generation and editing tasks to 2D image editing technologies and generates 3D objects through rapid 3D reconstruction, allowing users to iteratively refine the desired 3D objects through a combination of 2D editing and 3D reconstruction steps. The process is shown in Figure 1: Assume the users have a front-view image of a dog. First, they edit the front view using image editing methods to generate space glasses and a dashboard seamlessly into the scene. Next, employing multi-view diffusion technology, they can generate a back view. Then they edit the back-view image with the image editing methods again to add the backpack. Finally, the edited front and back images are input into a Dual-sided LRM model to generate a 3D model of the space dog. The entire process allows for step-by-step editing and completes each step within seconds, providing great convenience for rapidly editing the required 3D objects. This step-by-step method provides more precise control than end-to-end editing, enabling specific adjustments to image textures before reconstruction. Additionally, separately editing front and back views allows for more detailed customization.

Our proposed Dual-sided LRM, used in the final step of Tailor3D, generates 3D objects by receiving front and back images. As shown in the lower part of Figure 1, Having information from both sides allows for a more comprehensive understanding of the object, but it may lead to View inconsistency, referring to differences in geometry, color, and brightness in images taken from various angles and conditions, which can affect the quality of reconstruction. We extends LRM's capability from single-view to dual-view input, effectively handling inconsistencies between views. We introduce the LoRA Triplane Transformer Hu et al. (2022), which fine-tunes the LRM model with minimal memory consumption on a small dataset of 20K images to generate triplane features for both front and back views. This approach efficiently produces accurate triplane features, providing a solid foundation for subsequent feature fusion. Instead of merely stitching 2D image features, we combine the 3D triplane features of both views within 3D space. By applying Viewpoint Cross-Attention on the triplane, we merge these features swiftly, enhancing the quality of the final 3D

object. Additionally, we use data augmentation during training to further improve the model's robustness. Experimental results demonstrate that it excels in various 3D editing tasks, including geometric fill, texture synthesis, and style transfer.

Our contributions can be summarized as follows:

1. We propose Tailor3D, a rapid 3D editing pipeline. By combining 2D image editing and rapid 3D reconstruction techniques, it significantly enhances the efficiency of 3D editing.

2. Our Dual-sided LRM, combined with the LoRA Triplane Transformer, efficiently handles inconsistencies between front and back views, improving the overall reconstruction quality.

3. Tailor3D excels in various 3D editing and customization, particularly in local 3D generative fill, overall style transfer, and style fusion for objects, showcasing immense practical utility.

## 2 RELATED WORK

**Multi-view Diffusion for Objects.** Utilizing a single front-view image, multi-view diffusion demonstrates remarkable capabilities in synthesizing images from alternate viewpoints of the object Liu et al. (2024b); Shi et al. (2023a); Kong et al. (2024); Liu et al. (2024c); Tang et al. (2024b); Shi et al. (2023b); Wang & Shi (2023). These synthesized images are pivotal for subsequent stages of 3D object reconstruction to generate a mesh. Early efforts in this domain faced hurdles, particularly with small-scale training data and the imperative to ensure generalization performance Watson et al. (2023); Zhou & Tulsiani (2023); Chan et al. (2023); Szymanowicz et al. (2023); Wu et al. (2024); Fang et al. (2024). The improvement journey began with Zero-1-to-3 Liu et al. (2024b) refining Stable Diffusion Rombach et al. (2022) with the extrinsic camera parameters, marking a significant step in generalized multi-view diffusion. However, geometric consistency remained a challenge. SyncDreamer Liu et al. (2024c) built upon Zero-1-to-3, introducing a 3D-aware feature attention mechanism for enhanced synchronization, yielding 16 highly coherent multi-view images. Recent large models prefer using fewer overlapping canonical views (e.g., front, back, left, right) as inputs. This trend has led to the emergence of fixed-camera-parameter multi-view diffusion, simplifying training and enhancing multi-view consistency. For example, MVDream Shi et al. (2023b) and ImageDream Wang & Shi (2023) efficiently generate these four views, while zero123++ Shi et al. (2023a) extends this to six fixed views. Tailor3D improves practical utility by generating only the back image from the front, effectively addressing imperfect consistencies in diverse input scenarios.

**Large Model for 3D Reconstruction and Generation.** Early 3D generation methods initially focused on optimizing individual objects separately. SDS-based approaches Poole et al. (2023); Xu et al. (2023); Lin et al. (2023); Melas-Kyriazi et al. (2023); Wang et al. (2023a); Raj et al. (2023); Chen et al. (2023a); Tang et al. (2023); Wang et al. (2023b); Zhu & Zhuang (2024); Liang et al. (2023) utilized multi-view images from Zero-1-to-3 for this purpose. Subsequently, Diffusion + Reconstruction methods Liu et al. (2024a; 2023); Chen et al. (2024a); Long et al. (2024) expanded on SyncDreamer to optimize higher-consistency multi-view images. With the Large Reconstruction Model (LRM) scaling up in data and model size, it rapidly generates high-quality NeRF from single images in under 5s. This led to a shift where 2D methods handled generation tasks, and LRM managed 3D reconstruction. Consequently, 3D stable diffusion methods with fewer views, like MVDream Shi et al. (2023b), became preferred. For instance, Instant3D Li et al. (2024) uses 2D stable diffusion for four-view generation followed by LRM-like reconstruction. Similarly, LGM Tang et al. (2024a) and GRM Xu et al. (2024a) use Gaussian Splatting for reconstruction. For extensive 3D editing, we reduce perspectives to front and back, requiring lower consistency.

**3D Object Editing.** In 3D object domain, "customized editing" involves shape alterations, pattern addition, and texture application under user control. Traditional methods include explicit geometric representation editing, such as mesh deformation Yuan et al. (2021); Sorkine (2005); Sorkine & Alexa (2007), proxy-driven deformation Jacobson et al. (2012); Magnenat et al. (1988); Sederberg & Parry (1986); Yifan et al. (2020); Sumner et al. (2005); Gao et al. (2016), and data-driven deformation Gao et al. (2019; 2016), which utilize prior shapes for realistic outcomes. Over time, editing has moved towards implicit radiance fields Liu et al. (2019); Tan et al. (2018); Xu et al. (2021), especially on NeRFs Liu et al. (2021); Yang et al. (2021); Yuan et al. (2022). Earlier works focused on specific objects or scenes, lacking generalization Qi et al. (2024). In the 3D-AIGC era, 3D editing has evolved towards 2D image editing, reconstructed to generate new 3D objects Chen et al. (2023b;

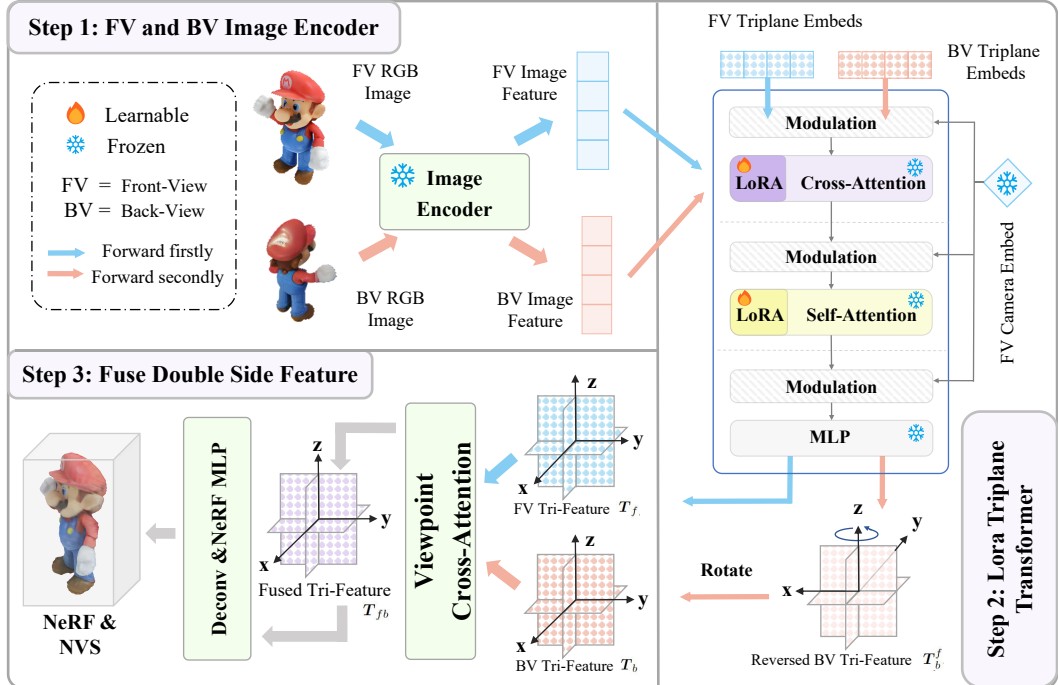

Figure 2: **Model Architecture of Dual-sided LRM**. We start with front and back view images. Then, using LoRA Triplane Transformer, we obtain front and back triplanes. Finally, we 'tailor' the two triplane features through rotation and Viewpoint Cross-Attention to obtain the 3D object.

2024b). MVEdit Chen et al. (2024b) denoises multi-view images and outputs high-quality textured meshes. However, its inference process takes 2-5 minutes, lacking real-time editing. In contrast, Tailor3D uses dual-side LRM to process inputs from both object sides, completing each editing step within seconds, enabling interactive 3D object editing.

## 3 METHODOLOGY

In this section, we present the pipeline and model architecture of Tailor3D. Firstly, we introduce the Large Reconstruction Model (LRM) and multi-view diffusion in Section 3.1. Next, in Section 3.2, we outline Tailor3D's process, illustrating 2D editing and rapid reconstruction into 3D objects. In Section 3.3, we delve into the Dual-sided LRM, accommodating inputs from imperfect consistent front and back views. We explain how the LoRA Triplane Transformer reduces memory usage and Viewpoint Cross-Attention to fuse 3D Triplanes from front and back views.

### 3.1 PRELIMINARIES

**Large Reconstruction Model (LRM).** LRM enables direct single-view to 3D reconstruction. The input image $I$ is encoded by an image encoder, producing patch-wise feature tokens $F \in \mathbb{R}^{N \times d_E}$, where $N$ is the number of image feature patches and $d_E$ is the dimension of the image encoder. Initial learnable positional embeddings for the triplane are defined as $f^{init}$ and engage in cross-attention with the image features $F$. They are modulated by the corresponding camera extrinsic parameters $E$ to generate the triplane feature map $T$.

$$T = (T_{xy}, T_{yz}, T_{xz}) = \text{TRI-FORMER}(f^{init}, F, E). \tag{1}$$

Here, $f^{init} \in (3 \times 32 \times 32) \times d_D$, where $d_D$ is the hidden dimension of the transformer decoder. TRI-FORMER incorporates self-attention, cross-attention, and modulation. The resultant triplane feature map $T \in (3 \times 64 \times 64) \times d_T$ comprises three planes: $T_{XY}$, $T_{YZ}$, and $T_{XZ}$. Resolution increases from 32×32 to 64×64 via deconvolutional layers. Finally, it undergoes $\text{MLP}^{nerf}$ for color and density derivation in NeRF rendering.

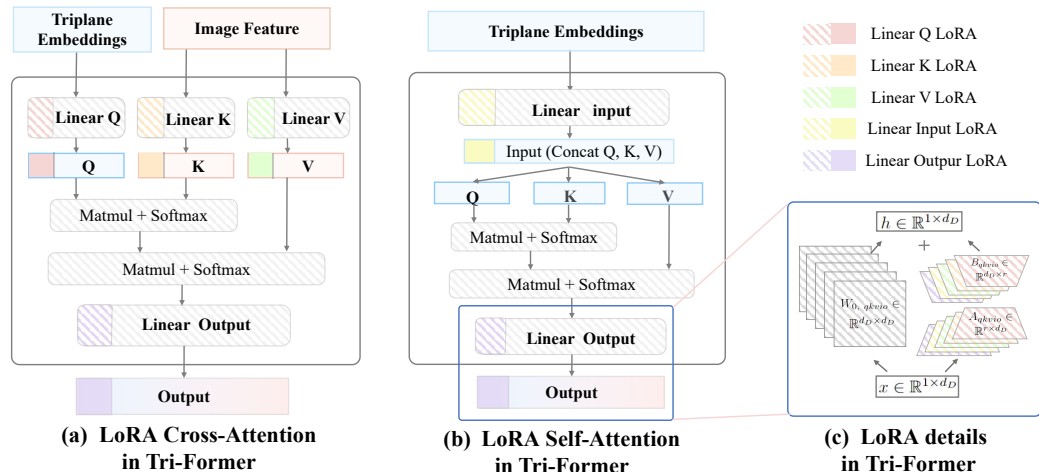

(a) **LoRA Cross-Attention**
**in Tri-Former**

(b) **LoRA Self-Attention**
**in Tri-Former**

(c) **LoRA details**
**in Tri-Former**

Figure 3: **LoRA Triplane Transformer**. (a) For Cross-Attention, we use the LoRA structure to replace the connection layers of $qkv$ and $output$. (b) For Self-Attention, we replace the connection layers of $input$ and $output$. Details of the LoRA are shown in (c).

**2D and Multi-view Diffusion.** The diffusion model iteratively denoises pure noise $x_T \sim \mathcal{N}(\mathbf{0}, \mathbf{I})$ over $T$ steps to yield clean data $x_0$, optimizing towards the gradient direction of the log probability distribution of the data, $\nabla_{\mathbf{x_t}} \log p(\mathbf{x_t})$. At step $t$, given the noisy input $x_t$, a neural network $\epsilon_\phi$ with parameters $\phi$ predicts the noise $\epsilon$.

$$\mathcal{L}_{diff}(\phi, x) = \mathbb{E}_{t,\epsilon}[\| \epsilon_\phi(x_t, t) - \epsilon \|_2^2]. \tag{2}$$

Multi-view diffusion generates images from specific objects based on current and desired viewpoints. By providing current image $\boldsymbol{I}$, extrinsic camera parameters $\boldsymbol{E} \in 4 \times 4$, alongside desired parameters camera $\boldsymbol{E}_o$, multi-view diffusion generates the image $\boldsymbol{I}_o$ for the desired viewpoint. In our pipeline, we utilize multi-view diffusion to generate the back image based on the front.

## 3.2 The Pipeline of Tailor3D

This section outlines Tailor3D's pipeline, as shown in the lower part of Figure 1. It begins with a front-facing image $\boldsymbol{I}_f$ of an object. Initially, image editing and style transfer are applied to create $\boldsymbol{I}_f'$. Next, multi-view diffusion methods like Zero-1-to-3 Liu et al. (2024b) generate the corresponding back image $\boldsymbol{I}_b$, which is then edited to get $\boldsymbol{I}_b'$. Finally, both $\boldsymbol{I}_f'$ and $\boldsymbol{I}_b'$ are input into Dual-sided LRM to obtain the final 3D object. Tailor3D offers various choices and potential variations. Original images $\boldsymbol{I}_f$ and $\boldsymbol{I}_b$ can be directly input into Dual-sided LRM for rapid reconstruction of the 3D object. Additionally, the back image $\boldsymbol{I}_b$ can be generated not only through Zero-1-to-3 but also through photography or direct provision. We will further elaborate on downstream tasks in the experimental section. The flexibility of Tailor3D arises from improved choices at each step and the robustness of our model, Dual-sided LRM, in handling imperfect consistency between front and back image inputs.

## 3.3 Dual-sided LRM: How to Accept Imperfect Consistent Views

In Section 3.2, our focus is on acquiring the edited front image $\boldsymbol{I}_f'$ and back image $\boldsymbol{I}_b'$ for an object. However, these images may exhibit imperfect consistency: They might not directly face the object, and their relationship can vary. Therefore, we need a reconstruction model capable of handling imperfectly consistent input images from both views to generate 3D objects. We select two views instead of four to reduce inconsistency pressure on editing and reconstruction. We explicitly merge two triplane features in the 3D domain, aiming to resolve the inconsistency issue intuitively.

**LoRA Triplane Transformer.** When employing pre-trained LRM parameters Hong et al. (2024), our goal is to minimize memory usage. In LRM, the single view feature $\boldsymbol{F}_f'$ is processed by a triplane transformer serving as a decoder to generate triplane NeRF features $\boldsymbol{T}_f$. This component facilitates mapping from a single view to 3D, enabling the model to understand diverse object shapes and infer

object information effectively. To minimize memory usage, we integrate the LoRA structure into the triplane transformer, as depicted in Figure 3. For self-attention, where $qkv$ is generated by shared linear layers, we replace all input and output linear layers with LoRA structures Hu et al. (2022). For cross-attention, where $qkv$ is generated by different linear layers, we replace all $qkv$ and output linear layers with LoRA structures. Specific details are as follows:

$$h^i = W_0^i x + \Delta W_{tp}^i x = W_0^i x + B_{tp}^i A_{tp}^i x. \tag{3}$$

Here, $i$ denotes the $i$-th Transformer layer. For self-attention, $tp$ represents the linear projection for *input* and *output*. For cross-attention, $tp$ denotes the linear projections for $q, k, v$, and *output*.

As shown in Figure 2, LRM generates the triplane feature $\boldsymbol{T}_f$ for the front view from features $\boldsymbol{F}'_f$ and camera parameters $\boldsymbol{E}_f$. Similarly, for the back view features $\boldsymbol{F}'_b$, we use the camera parameters $\boldsymbol{E}_f$ of the front view to obtain the triplane feature $\boldsymbol{T}_b^f$ for the back view through the LoRA triplane transformer, as expressed by the following equation:

$$\boldsymbol{T}_f / \boldsymbol{T}_b^f = \text{TRI-FORMER}_{\text{LoRA}}(\boldsymbol{f}^{init}, \ \boldsymbol{F}'_f / \boldsymbol{F}'_b, \ \boldsymbol{E}_f). \tag{4}$$

Here $\boldsymbol{T}_b^f$, the triplane feature for the back view obtained using the front view's camera parameters, cannot be directly merged with $\boldsymbol{T}_f$. We will address this and the inconsistency between the front and back view angles in the next section.

**Fuse Double Side Feature.** To merge the two triplane features $\boldsymbol{T}_f$ and $\boldsymbol{T}_b^f$, we first horizontally flip $\boldsymbol{T}_b^f$ by 180 degrees around the z-axis to obtain $\boldsymbol{T}_b$. Due to inconsistency between the front and back views, direct alignment or addition of the triplane features isn't feasible. Leveraging the triplane representation, we apply Viewpoint Cross-Attention to each plane individually. We use $\boldsymbol{T}_f$ as the query and $\boldsymbol{T}_b$ as the key and value to incorporate missing information from the backside. We adopt a window-based attention structure, with a window size set to 7, significantly reducing memory consumption. This yields the final $\boldsymbol{T}_{fb}$, encapsulating information from both views. Data augmentation further bolsters robustness to inconsistency, with back view images undergoing scaling, rotation, and translation, each with a 10% probability.

Finally, the Triplane-NeRF formulation utilizes $\text{MLP}^{nerf}$ to derive NeRF color and density parameters for volume rendering. Supervision includes $V$ views, comprising the front, back and $(V-2)$ randomly chosen side views. For a specific view $v$, the loss function for synthesizing the prediction $\hat{x}_v$ and the ground truth $\boldsymbol{x}_v^{GT}$ for new view composition is formulated as follows:

$$\mathcal{L}(\boldsymbol{x}) = \frac{1}{V} \sum_{v=1}^{V} \left( \lambda_1 \mathcal{L}_{\text{MSE}}(\hat{\boldsymbol{x}}_v, \boldsymbol{x}_v^{GT}) + \lambda_2 \mathcal{L}_{\text{LPIPS}}(\hat{\boldsymbol{x}}_v, \boldsymbol{x}_v^{GT}) + \lambda_3 \mathcal{L}_{\text{TV}}(\hat{\boldsymbol{x}}_v, \boldsymbol{x}_v^{GT}) \right). \tag{5}$$

$\mathcal{L}_{\text{MSE}}$ denotes the normalized pixel-wise L2 loss, $\mathcal{L}_{\text{LPIPS}}$ is perceptual image patch similarity. $\mathcal{L}_{\text{TV}}$ is the total variation loss to prevent noise in the image. Weight coefficients $\lambda_1, \lambda_2, \lambda_3$ are applied.

## 4 EXPERIMENTS

This section explores the experimental aspects. In Section 4.1, we delve into various implementation details, including dataset, model architecture parameters, camera adjustments, and training/testing processes. In Section 4.2, we present experimental results. We showcase Tailor3D's versatility across different tasks and conduct ablation studies on key modules.

### 4.1 IMPLEMENTATION DETAILS

For the dataset, LRM pre-trained weights Hong et al. (2024); He & Wang (2023) are trained on Objaverse Deitke et al. (2023), containing 730K objects rendered from 32 random viewpoints. Fine-tuning uses 22K high-quality 3D objects from the Gobjaverse-LVIS Qiu et al. (2023); Gupta et al. (2019) dataset. Training involves front and back views, plus random side views for new view synthesis. More details about the dataset are shown in the Appendix C.2 of the appendix.

We use the network architecture from the pre-trained LRM model. The image encoder is based on DINOv2's ViT-B/16 model Oquab et al. (2023), operating at a resolution of $384 \times 384$. The image

features have a dimensionality of 768. The triplane transformer decoder consists of 16 layers with 16 transformer heads, featuring positional embeddings of dimensionality 1024 and triplanes with dimensionality 80. $\mathrm{MLP}^{nerf}$ comprises 10 layers. We set the LoRA rank to 4 for the LoRA Triplane Transformer. During neural rendering, we sample 128 points along each ray and produce images at a resolution of $128{\times}128$. For camera normalization, we align with LRM standards, positioning the camera at $[0, -2, 0]$ relative to the object center. This ensures the object's z-axis is upward, and the front view corresponds to the negative y-axis. External rendering parameters are normalized relative to the reference view. We train for 10 epochs on 8 A100 GPUs with a batch size of 16, taking about 6 hours. The loss function coefficients are $\lambda_1 = \lambda_2 = \lambda_3 = 1.0$. We use the AdamW optimizer with a learning rate of $3{\times}10^{-4}$ and a cosine schedule. During inference, we query a resolution of $384{\times}384{\times}384$ points from the reconstructed triplane-NeRF, completing it in less than 5 seconds.

## 4.2 EXPERIMENT RESULTS

In Section 4.2.1, we showcased Tailor3D's capabilities in 3D generation, covering geometric object fill, texture synthesis, and style transfer. In Section 4.2.2 and Section 4.2.3, we compared our approach with existing 3D Generation methods and multi-view editing methods. In Section 4.2.4, we performed ablation experiments and finally we show some failure cases in Section 4.2.5.

### 4.2.1 TAILOR3D APPLICATIONS

We showcase its versatility in 3D Generative Geometry / Pattern Fill, encompassing local geometric shape and texture pattern filling. We highlight its style transfer and fusion capabilities, allowing for operations like style transfer and blending two styles onto one object. Tailor3D enables users to edit both the front and back of objects, expanding editing possibilities for customized 3D objects.

**3D Generative Geometry / Pattern Fill.** Here, we showcase Tailor3D's local 3D object filling ability, as depicted in Figure 4. Demonstrating step-by-step object filling and editing through text or image prompts. In Row 2, starting from armor, we generate a medieval general by adding the head, hands, and cloak progressively. Row 3 illustrates additional object manipulation, including the addition of a mailbox, balloons, a flower bush, and a basketball hoop.

**3D Style Transfer and Fusion.** Tailor3D also demonstrates its transfer and fusion capabilities for various styles. Unlike previous approaches, Tailor3D ensures IP integrity while offering flexibility in specifying styles through images or text guidance. Notably, it leverages Midjourney for 2D image generation and editing. Additionally, Tailor3D enables the infusion of different styles onto both the front and back of objects, showcasing the effectiveness of the Dual-sided LRM's merging ability.

### 4.2.2 COMPARE TO EXISTING 3D IMAGE-TO-3D GENERATION METHODS

We compare our approach with Wonder3D Long et al. (2024), TriplaneGaussian Zou et al. (2023), and LGM Tang et al. (2024a) on a test set of 100 images generated by stable diffusion Rombach et al. (2022). Each model takes a single image as input and generates multiple views using multi-view diffusion, while our method only generates an additional back view. Quantitative results are provided in Table 1 alongside generation times, highlighting the practical value of our method. Quantitative results are shown in the Figure 16 of the appendix.

### 4.2.3 COMPARED TO MULTI-VIEW EDITING METHODS.

Here, we compare Tailor3D with multi-view editing methods like MVEdit Chen et al. (2024b); Haque et al. (2023). Existing multi-view approaches are optimization-based, requiring separate optimization for each object or scene and re-optimization for every edit. In contrast, Tailor3D uses a feed-forward framework, completing reconstruction in under 5 seconds. Multi-view methods can only be controlled via text and can edit only the front side of a 3D object, lacking precision for local edits and maintaining object identity. Tailor3D, however, supports text or image-based instructions for both global and local edits, as shown in Figure 8. It can edit Mario's overall style while preserving identity, which MVEdit cannot, and it can also modify local parts.

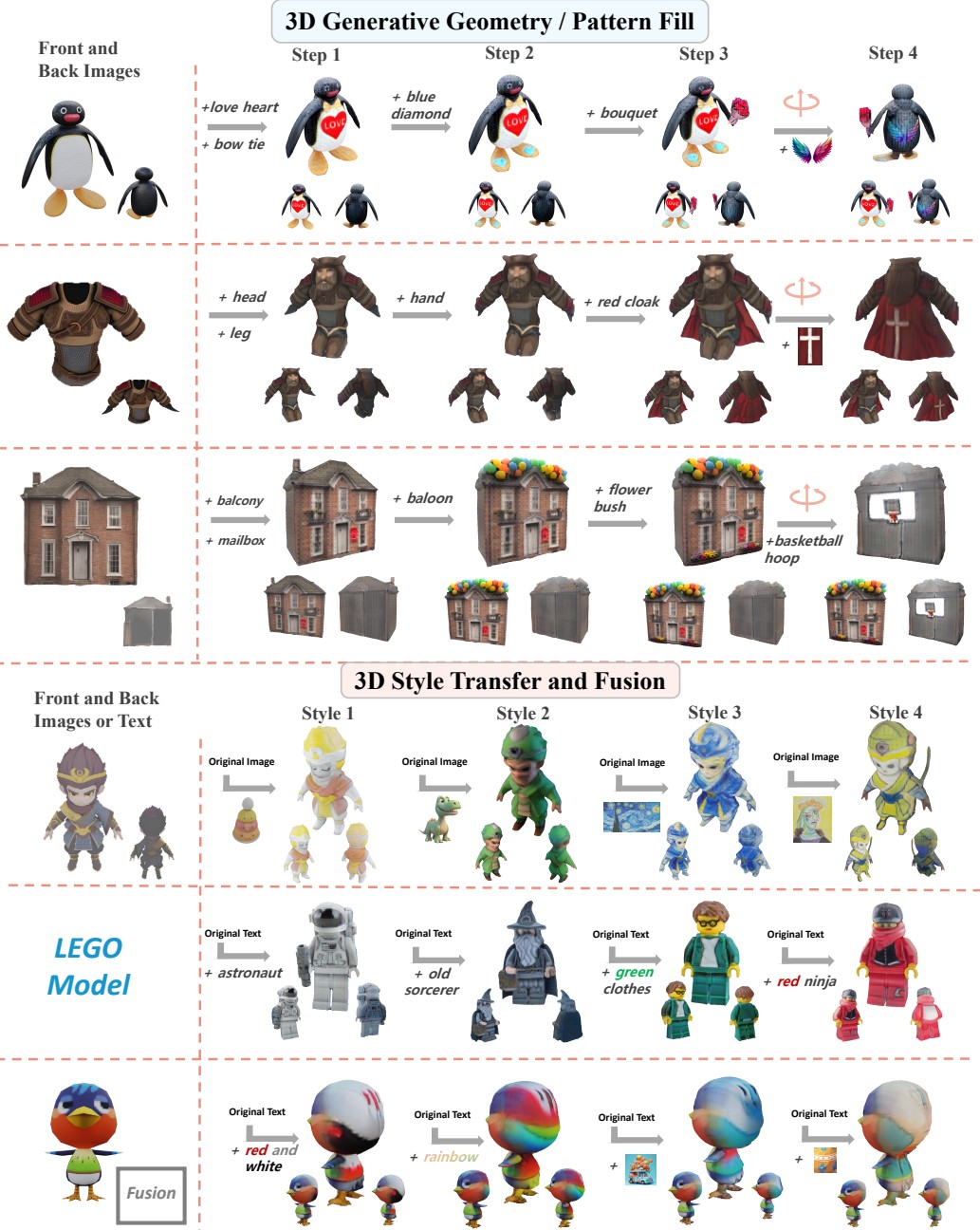

Figure 4: **3D Generative Fill and 3D Style Transfer.** It includes both Geometry Fill and Pattern Fill, allowing us to add or modify local geometric structures or texture patterns of 3D objects. Guidance can be provided through text or images as prompts. Additionally, we offer style images or textual guidance to transform 3D objects into desired styles. Ensuring the maintenance of IP integrity during disguise adds significant practical value to 3D tasks.

### 4.2.4 ABLATION STUDY

We perform an ablation study on the Dual-sided LRM, focusing on three aspects: the fusion of 3D features from both sides, the rank of the LoRA Transformer, and the extrinsic camera parameters of front and back images. Results are presented in Table 2, using the same test set as in Section 4.2.2.

**The Way to Fuse Double Side Feature.** We use Viewpoint Cross-Attention to fuse features from two sides, and also experiment with 2D conv layers and direct addition. As shown in Table 2(a), Viewpoint Cross-Attention achieves the best results. Figure 5 provides qualitative results on a bird example, demonstrating its effectively stitches the front and back sides together.

Table 1: **Comparison with Existing 3D Generation Methods.** We compare single image-to-3D methods, including common metrics and user studies. Results indicate that ours outperforms others.

| Compare with others. | | Common Metrics | | | User Study ↑ (0 to 100 score) | | |
| --- | --- | --- | --- | --- | --- | --- | --- |
| Methods | InF. Time. | LPIPS ↓ | SSIM ↑ | PSNR ↑ | Geometry | Texture | Overall |
| TriplaneGaussian | 20s | 0.2811 | 0.5635 | 14.89 | 56.3 | 54.5 | 62.3 |
| Wonder3D | 3min | 0.2709 | 0.6485 | 16.23 | 73.3 | 76.3 | 79.2 |
| LGM | 5s | 0.2473 | 0.8423 | 19.02 | 79.3 | **85.2** | 83.2 |
| Tailor3D (Ours) | 5s | **0.2345** | **0.8525** | **19.34** | **82.3** | 84.2 | **86.3** |

Table 2: **Abalation Study.** We conducted ablation regarding the fusion method for both sides, the rank of the LoRA Triplane Transformer, and the extrinsic camera parameters. †: VP-CA means Viewpoint Cross-Attention. ∗: The first is the front-view extrinsic and the second is for the back.

**(a) Way to Fuse Double Sides.**

| Fuse Way | Score | SSIM↑ | LPIPS↓ |
| --- | --- | --- | --- |
| Add | 76.3 | 0.7377 | 0.2938 |
| Conv2D | 84.2 | 0.8239 | 0.2443 |
| VP-CA† | **86.3** | **0.8525** | **0.2345** |

**(b) LoRA Transformer Rank.**

| Rank | Score | SSIM↑ | LPIPS↓ |
| --- | --- | --- | --- |
| 2 | 79.2 | 0.7623 | 0.2877 |
| 4 | **86.3** | **0.8525** | **0.2345** |
| 8 | 82.2 | 0.7902 | 0.2535 |

**(c) Two Camera Extrinsics.**

| Cam Ext. ∗ | Score | SSIM↑ | LPIPS↓ |
| --- | --- | --- | --- |
| $E_b + E_b$ | 60.5 | 0.6288 | 0.3944 |
| $E_f + E_b$ | 33.4 | 0.3523 | 0.5653 |
| $E_f + E_f$ | **86.3** | **0.8525** | **0.2345** |

Figure 5: **Way to Fuse Double Sides.** VP-CA achieves the best results to fuse them together.

Figure 6: **Change the Geometry of the Back.** Currently difficult to change from the back.

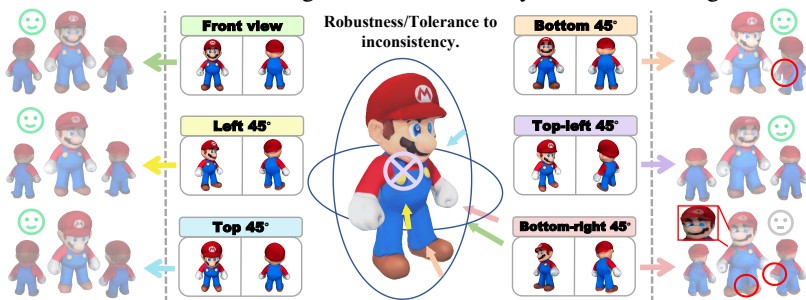

Figure 7: **Robustness to Handle Inconsistency.** It does not require defining front and back sides due to its robustness to inputs from various directions.

**The Rank of LoRA Triplane Transformer.** We conduct ablation experiments on the rank of the LoRA Triplane Transformer, setting the rank to 2, 4, and 8, respectively. Our experimental results indicate that a rank of 4 achieves the best performance.

**Extrinsic Camera Parameters.** We apply the same front camera parameters $E_f$ to both front and back images, rotating only the back triplane. We also experiment with separate camera parameters, $E_f$ and $E_b$, without rotation. Results show that using only front extrinsics provides accurate outcomes, as the LRM structure accepts only front camera parameters.

**Change the geometry of the back side.** Our geometric editing is limited to the front view, while for the back, we mainly edit patterns in a central area. In Figure 6, we show an example of adding wings to a penguin's back, which is possible within the back area, but adding objects like a volleyball outside is not. Structural changes are usually made from the front, as seen in the third row where we added a volleyball. We plan to support multi-view geometric changes in the future version.

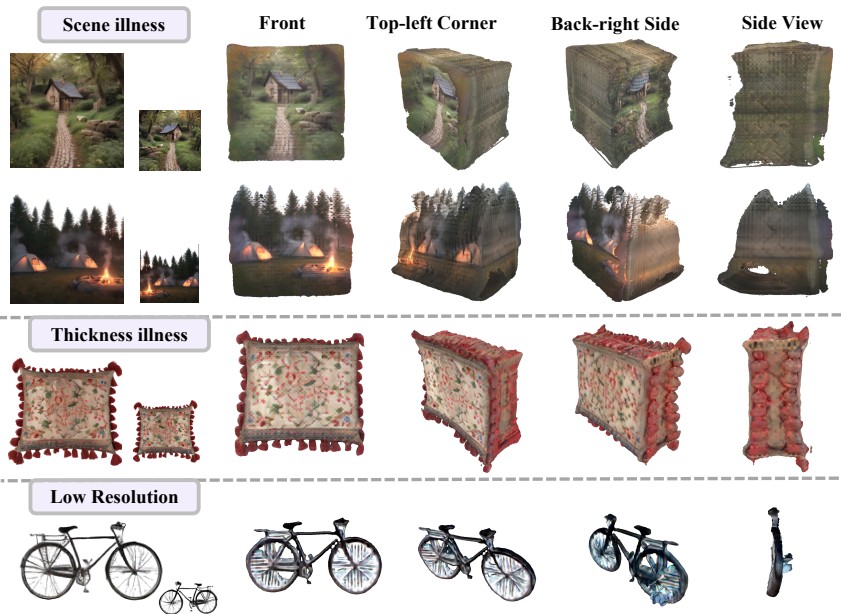

Figure 8: **Compared to Multi-View editing methods (MVEdit).** Tailor3D can accept both text and image guides, and the editing process can maintain the object's identity and geometry.

Figure 9: **Failure case (Two-view Reconsctruction).** We provide front and back views for reconstruction, showing its poor performance in micro-scenes, thickness estimation and low resolution.

**Tailor3D's robustness to handle inconsistency.** We didn't strictly define the front and back orientation because Tailor3D handles inconsistencies well. As shown in Figure 7, tests with Mario images from various non-strict front and back views demonstrate that Tailor3D tolerates inconsistency, successfully reconstructing the 3D object despite detail variations from different angles.

### 4.2.5 FAILURE CASE (TWO-VIEW RECONSCTRUCTION)

We present additional failure cases. Without editing, we simply provide the front and back views for reconstruction. Figure 9 highlights issues like poor performance in micro-scenes, inaccurate blanket thickness estimation, and low-resolution bicycle meshes. We plan to fix them in the future.

## 5 CONCLUSION

We introduce Tailor3D, a tool for quickly creating customized 3D assets using editable dual-sided images. By combining 2D editing and fast 3D reconstruction, users can iteratively refine objects. Our Dual-sided LRM and LoRA Triplane Transformer act as 'tailors,' stitching front and back views to handle inconsistencies and enhance reconstruction. Experiments show Tailor3D's effectiveness in tasks like 3D generative fill and style customization, providing a user-friendly, cost-effective solution for rapid 3D editing in animation, game development, and more.

**Code of Ethics/Reproducibility and Ethics statement.** There is no ethics about the paper and code and all code can be reproduced. All code will be public soon.

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

## A    APPENDIX OVERVIEW

We first introduce additional background information in the supplementary materials in Appendix B. We first divided 3D Reconstruction into three categories and introduced the LRM Hong et al. (2024) family. In Appendix C, we presented additional details regarding the methodology and implementation of experiments. We emphasize the differences between our training configuration and the original LRM and provide further insights into the Gobjaverse Qiu et al. (2023) dataset. In Appendix D, we mainly present the additional experimental results. We first present additional examples of Tailor, followed by comparisons with more multi-view reconstruction methods.

## B    ADDITIONAL RELATED WORK

This section categorizes 3D reconstruction into single-view reconstruction, multi-view reconstruction, and the recently popular normal-view reconstruction. We then delve into the benefits of employing double-sided information for canonical-view reconstruction in appendix B.1. Following that, we introduce articles from the LRM family Hong et al. (2024); Li et al. (2024); Wang et al. (2024); Xu et al. (2024b) in appendix B.2, discussing various variants of this framework.

### B.1    SINGLE, MULTI AND CANONICAL-VIEW RECONSTRUCTION

Firstly, we delineate several types of reconstruction. Single-view reconstruction involves generating a 3D mesh of an object from a single-view image (typically the front view). On the other hand, multi-view reconstruction typically involves multiple viewpoint images of an object along with corresponding camera extrinsic (often 20-100 views), aiming to reconstruct a 3D object. A landmark method in this domain is NeRF, which utilizes MLPs for novel view synthesis or 3D reconstruction. However, NeRF-based methods suffer from the need for individual optimization for each object, resulting in long reconstruction times, sometimes reaching 1-2 hours. Early 3D generation methods which use multi-view diffusion for generating multiple views of an object and subsequent reconstruction Liu et al. (2024a); Long et al. (2024), also face long reconstruction times.

The development trajectory of NeRF involves the need for increasingly fewer viewpoints for reconstruction, fewer camera parameters, and faster reconstruction speeds. However, these methods still require individual optimization for each object. In contrast, LRM serves as a universal reconstruction model. As the model and dataset sizes reach a particular scale, reconstruction models become universal, eliminating the need for individual optimization of objects to be reconstructed. Within this universal framework emerges a reconstruction method known as canonical-view reconstruction, which uses fixed faces for reconstruction, typically the front, back, left, and right faces, referred to as 4-canonical-view reconstruction. Instant3D Li et al. (2024), TriplaneGaussian Zou et al. (2023), and LGM Tang et al. (2024a) all employ this reconstruction method. However, the challenge with using the front, back, left, and right faces lies in effective editing, as it is difficult to edit all four faces simultaneously. Tailor3D adopts **Dual-Canonical-view Reconstruction**, utilizing only the front and back faces with fewer overlaps, facilitating user editing. Here, we emphasize that multi-view reconstruction requires optimization for individual objects, whereas canonical-view reconstruction is built upon a general reconstruction framework.

### B.2    INTRODUCTION TO LRM FAMILY

As mentioned earlier, early 3D generation methods utilized multi-view diffusion to generate additional viewpoints from a single image and optimized the multi-view reconstruction of a 3D object based on these views, which need **a few minutes**. The LRM family, serving as a series of Feed-Forward Methods, directly generates 3D meshes without the need to synthesize multiple viewpoint images or training and adapt to models such as NeRF **within only several seconds**. It represents a universal reconstruction framework. As illustrated in Figure 10, LRM is a universal framework for single-view reconstruction. That is, a single image can directly generate a 3D mesh. The fundamental concept involves predefining the feature map of Triplane NeRF and then performing cross-attention with 2D images and their corresponding camera parameters. The resulting feature map can directly provide novel views of images or even the entire 3D mesh in the Triplane NeRF format.

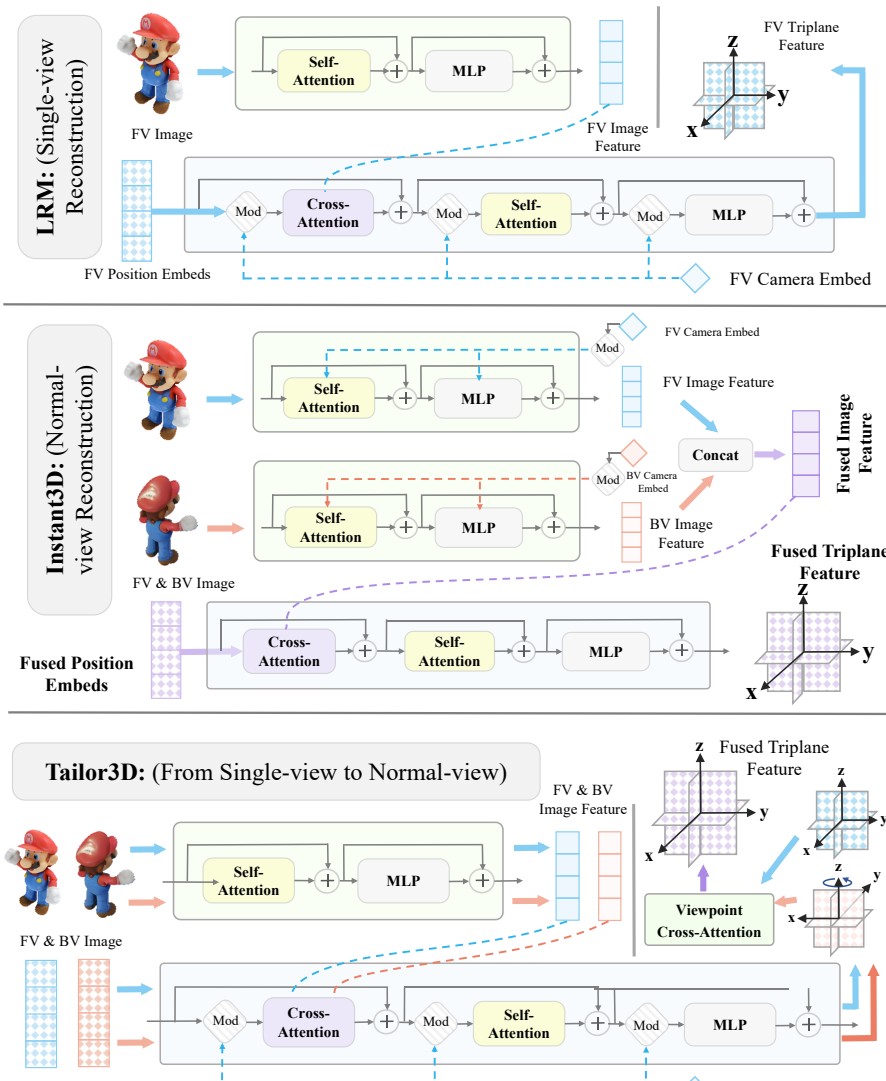

Figure 10: Model architectures of LRM, Instant3D and Tailor3D.

Building upon this foundation, Instant3D Li et al. (2024) addresses normal 4-canonical-view reconstruction. It involves two stages: first, utilizing a 2D diffusion model to obtain front, back, left, and right images of an object from text prompts; second, reconstructing the 3D object from these four viewpoints. PF-LRM Wang et al. (2024) focuses on pose-free sparse multi-view reconstruction, enabling the generation of a 3D object from three images taken from arbitrary viewpoints without corresponding camera extrinsics. However, its framework complexity arises from the supervision involving PnP and various geometric theories. DMV3D Xu et al. (2024b), an extension of Instant3D, introduces a denoising process, resulting in a denoised multi-view diffusion framework. Unfortunately, these methods have not been open-sourced yet, with only the OpenLRM He & Wang (2023) codebase providing the inference code for LRM.

LRM and Instant3D can be regarded as methods corresponding to single-view and 4-canonical-view reconstruction, respectively. However, their handling of camera parameters differs. As shown in fig. 10, LRM adjusts camera parameters with triplane features in the triplane transformer decoder. In practice, the external camera parameters are fixed, meaning the camera is positioned at $[0, -2m, 0]$ and oriented to look directly at the object along the positive y-axis. Hence, LRM can only accept the camera parameters of the front view, as demonstrated in Table 2c. In contrast, Instant3D places the modulation of the camera within the image encoder. After obtaining image features from four views, these features are concatenated and passed through the triplane transformer decoder. This approach

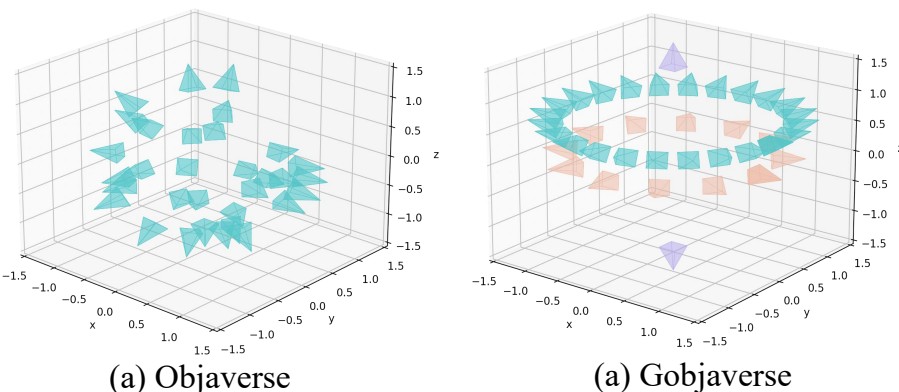

(a) Objaverse          (a) Gobjaverse

Figure 11: Rendering perspectives in Objaverse and Gobjaverse.

involves merging the features from multiple viewpoints at the 2D image feature level. However, this approach is not a natural transition from single-view to canonical-view reconstruction. We choose to utilize Viewpoint Cross-Attention to fuse the 3D triplane features of the front and back views. This allows us to easily extend single-view reconstruction to dual(4)-canonical-view reconstruction using only the pre-trained weights from the single-view reconstruction. Furthermore, only training the Viewpoint Cross-Attention is necessary to minimize costs.

## C  ADDITIONAL METHODOLOGY

In this section, we discuss the training and experimental aspects. In appendix C.1, we describe our training setup, using the LRM model from the OpenLRM codebase He & Wang (2023), and delineate the variations in the parameter quantities compared to the original LRM. In appendix C.2, we offer a detailed overview of viewpoint rendering in the Gobjaverse dataset Qiu et al. (2023). We achieved satisfactory results with a relatively small dataset size by utilizing meticulously crafted artificial rendering data that boast high-quality textures and excellent consistency (22K).

### C.1  TRAINING SETTINGS

Here, we focus on describing our training details. First, we utilized the OpenLRM codebase as the basis for our LRM implementation. The original resolution is 512, but we used 256. The dimensionality of the triplane feature map, which was initially 80, was reduced to 40. Other model parameters remain unchanged, such as the dimensionality of camera embeddings (1024) and triplane transformer (1024). We used 96 rendering sample rays. For training parameters, the learning rate was set to $3e - 4$, with a weight decay of 0.05. We employed a cosine scheduler. The total batch size was set to 16 (across 8 A100 GPUs), and we trained for a total of 20 epochs.

### C.2  DATASET: GOBJAVERSE

We utilized the Gobjaverse dataset Qiu et al. (2023), an enhanced version of the Objaverse dataset with higher-quality rendering. Unlike Objaverse, which renders a single object with randomly positioned cameras spherically, Gobjaverse performs orbit rendering around an object, capturing two orbits shown in Figure 11. In the higher-elevation orbit, 24 views at equal intervals are represented in cyan. In the lower-elevation orbit, 12 views at equal intervals are represented in red. Additionally, two views captured from the top and bottom are represented in purple.

We excluded the two views captured from the top and bottom during our training process. This allowed our training data to provide input from both the front and back sides of the objects. It is worth noting that the opposite directions are only along the x-axis and y-axis. In the z-axis direction, they have the same elevation angle rather than being utterly symmetric across the center. This approach differs from methods like Instant3D and LGM Tang et al. (2024a), which use techniques similar to MVDream Shi et al. (2023b) to generate 4 views of an object using 2D diffusion. Gobjaverse offers higher consistency, resulting in higher data quality, which facilitates the fusion of features from the front and back directions.

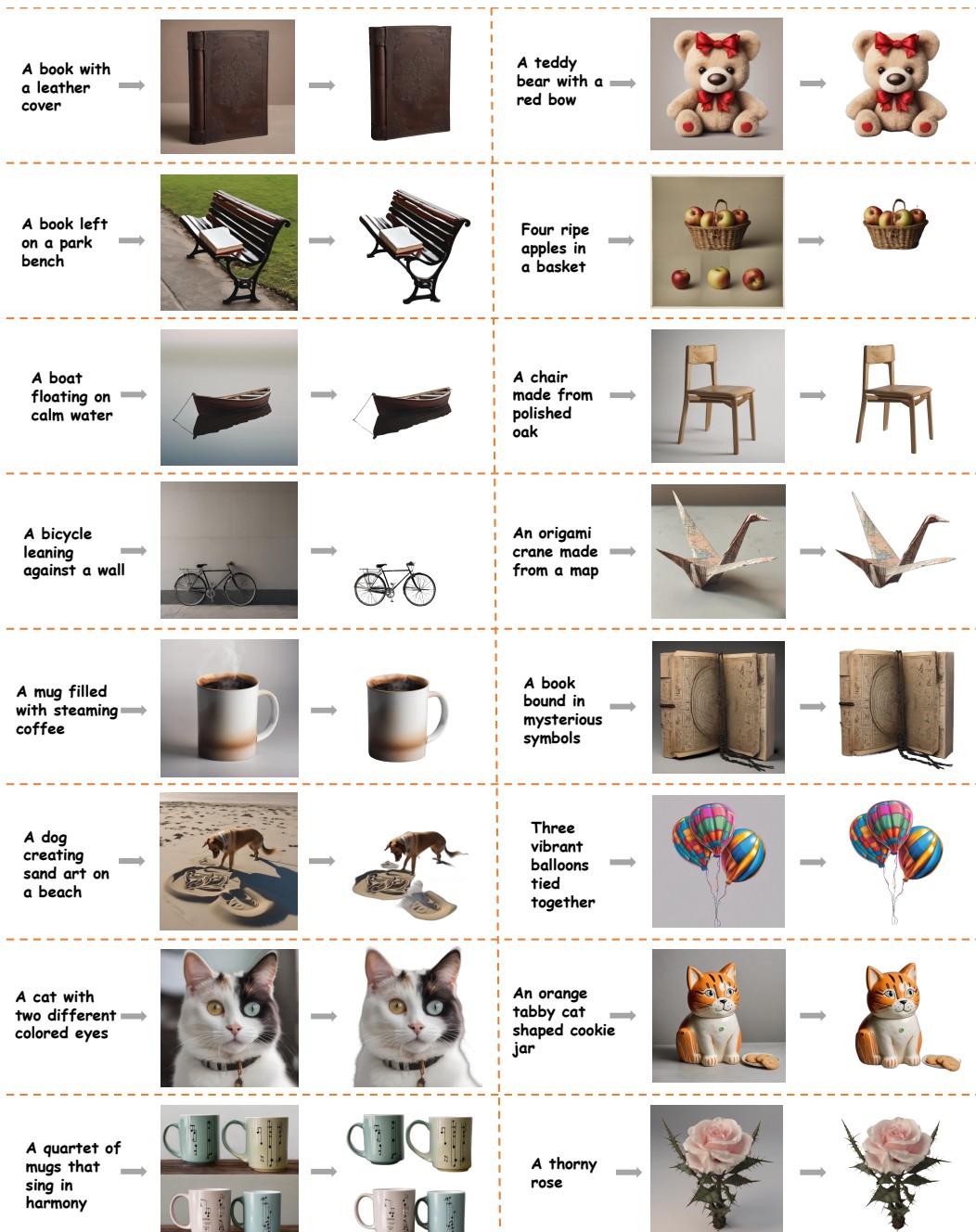

Figure 12: Testset: 100 3D Assets from Stable Diffusion (1).

## C.3 TESTSET: 100 IMAGES FROM STABLE DIFFUSION

Our quantitative test set and a portion of the qualitative test set consist of 100 objects generated by Stable Diffusion, with the background removed. Here, we present partial examples using two images, while the remaining qualitative examples may come from the use of Midjourney for generation. Our test set covers various objects and micro-scenes such as animals, humans, plants, and landscapes, enabling a comprehensive assessment of the quality of the generation models. Additionally, all our models comply with copyright and related regulations.

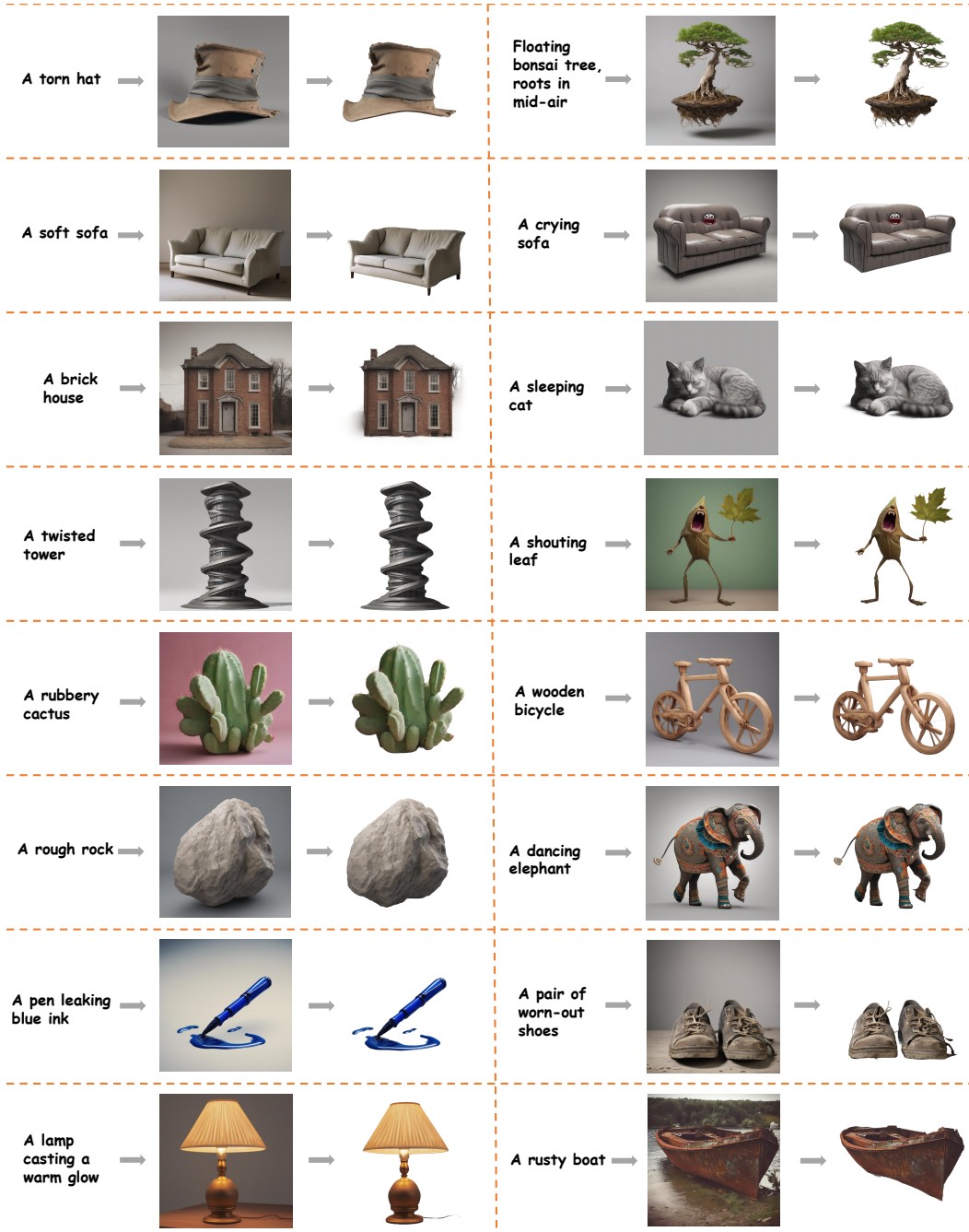

Figure 13: Testset: 100 3D Assets from Stable Diffusion (2).

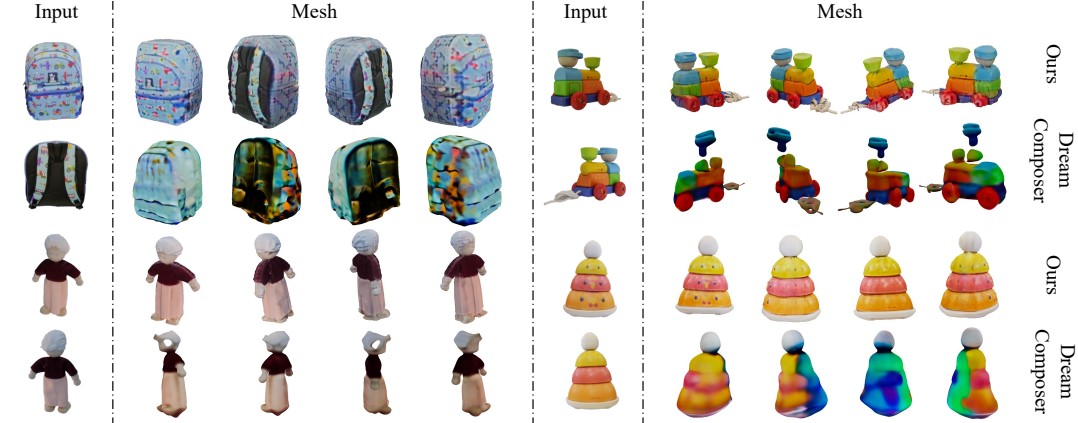

Figure 14: **Compare with Dreamcomposer.** Here, we present a comparison with the multi-view DreamComposer Yang et al. (2024). In this comparison, we provide Tailor3D with ground-truth RGB images for the back side. It can be observed that Tailor3D exhibits more detailed texture features and avoids defects such as holes.

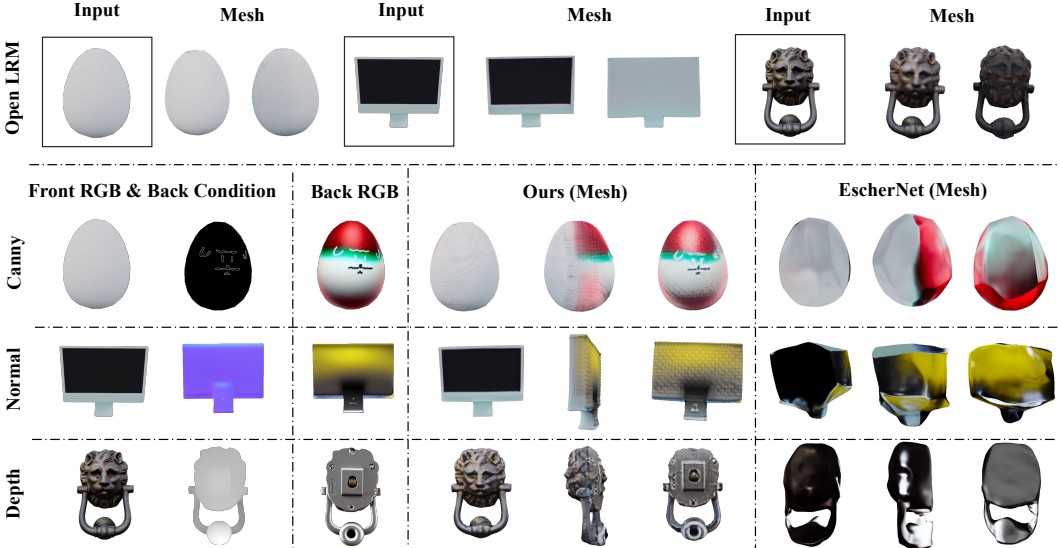

Figure 15: **Compare withh multi-view input model EscherNet Kong et al. (2024).** Our created mesh excels beyond other methods, delivering superior speed and quality.

# D  ADDITIONAL EXPERIMENTS

In this section, we show more experiments. In appendix D.1, we compare our method's effectiveness with more recent multi-view reconstruction techniques. In appendix D.2, similar to Figure 4, We present additional examples of Tailor3D, showcasing our ability to customize and edit objects.

## D.1  COMPARISON WITH MORE MULTI-VIEW RECONSTRUCTIONS

In the main paper, we compared earlier 3D generation methods like Wonder3D Long et al. (2024), TriplaneGaussian Zou et al. (2023), and LGM Tang et al. (2024a), most of which were focused on image-to-3D generation. In the main text, we provided only quantitative results, as shown in Table 1. Here, we present the qualitative results. Qualitative results. Figure 16 demonstrates Tailor3D's capability to enhance backside information with Dual-sided LRM. Wonder3D and TriplaneGaussian struggle with complex objects, exhibiting lower overall quality. LGM, using Gaussian representation, suffers from ghosting effects and lacks detail in features like tree leaves.

Conversely, approaches like Dreamcomposer Yang et al. (2024) and EscherNet Kong et al. (2024) aimed to complement additional viewpoints in the Table 1. It's worth noting here the test set is

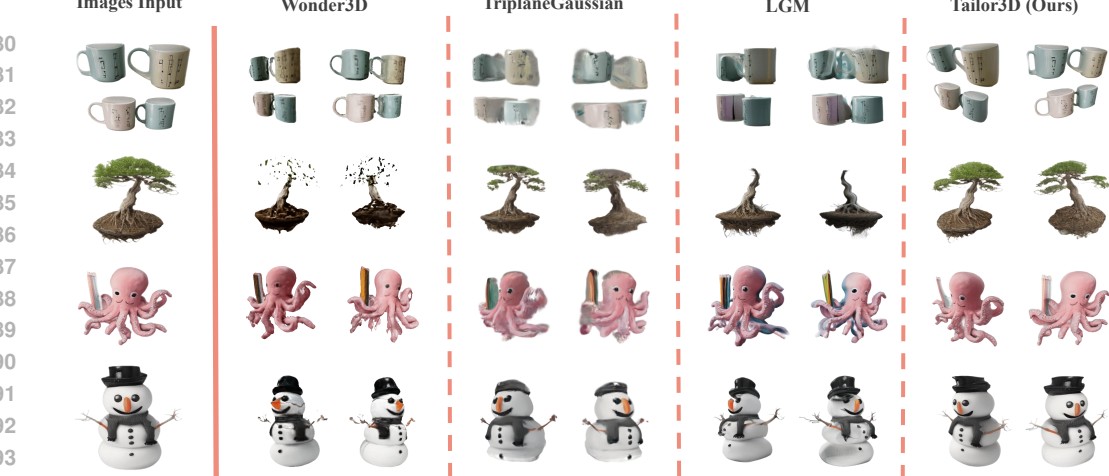

| Images Input | Wonder3D | TriplaneGaussian | LGM | Tailor3D (Ours) |

Figure 16: **Qualitative Results: Compare to Existing 3D Generation.** We compare single image-to-3D methods. Wonder3D and TriplaneGaussian have lower resolutions, while LGM often shows ghosting effects with complex textures. Our method, however, achieves superior results.

from GSO30 Downs et al. (2022) and Objaverse Deitke et al. (2023) datasets instead of the 100 SD test set used in the main paper. Dreamcomposer and EscherNet are optimization-based methods, thus requiring several minutes to generate 3D results. In contrast, Tailor3D only needs 5 seconds to produce superior 3D reconstruction results.

**Comparison with Dreamcomposer.** DreamComposer is built on SyncDreamer Liu et al. (2024c), allowing it to accept inputs from multiple viewpoints and fill in missing information for all sides except the back. In our experimental results (see fig. 14), we adjusted the back input to be the RGB image of the ground-truth back side for comparison purposes. That is, we provided Tailor3D and Dreamcomposer with pictures of the front and back of the object, which could have been more perfectly consistent. We found that Tailor can generate superior mesh results compared to Dream-Composer. DreamComposer tends to exhibit more defects in its reconstructions.

**Comparison with EscherNet.** EscherNet is a multi-view conditional diffusion model for viewpoint synthesis. It learns implicit and generative 3D representations combined with Camera Position Encoding (CaPE). EscherNet can generate more consistent images and has higher reconstruction quality. In this experiment, we provided EscherNet with 16 viewpoints, while our Tailor3D had only the front and back viewpoints. Even in this scenario, our approach still has a significant advantage and obtains better mesh results. This further demonstrates that our method using only two views for reconstruction can achieve better results.

## D.2 MORE EXAMPLES

Here, we showcase more qualitative examples, including 3D style transfer, style fusion, and 3D generative fill. We demonstrate the model's ability to transform overall styles as well as perform localized editing. These examples show the potential for industrial applications.

## E LIMITATIONS

Despite the strong performance of Tailor3D. However, relying solely on front and back views for object reconstruction may encounter challenges with objects of certain thicknesses. Additionally, the generated 3D object meshes may have lower resolutions, and the addition of geometric features may not significantly alter the mesh. We will further investigate methods to address the generation and reconstruction of objects with thicker side profiles in future work, aiming to enhance the quality and resolution of the meshes.

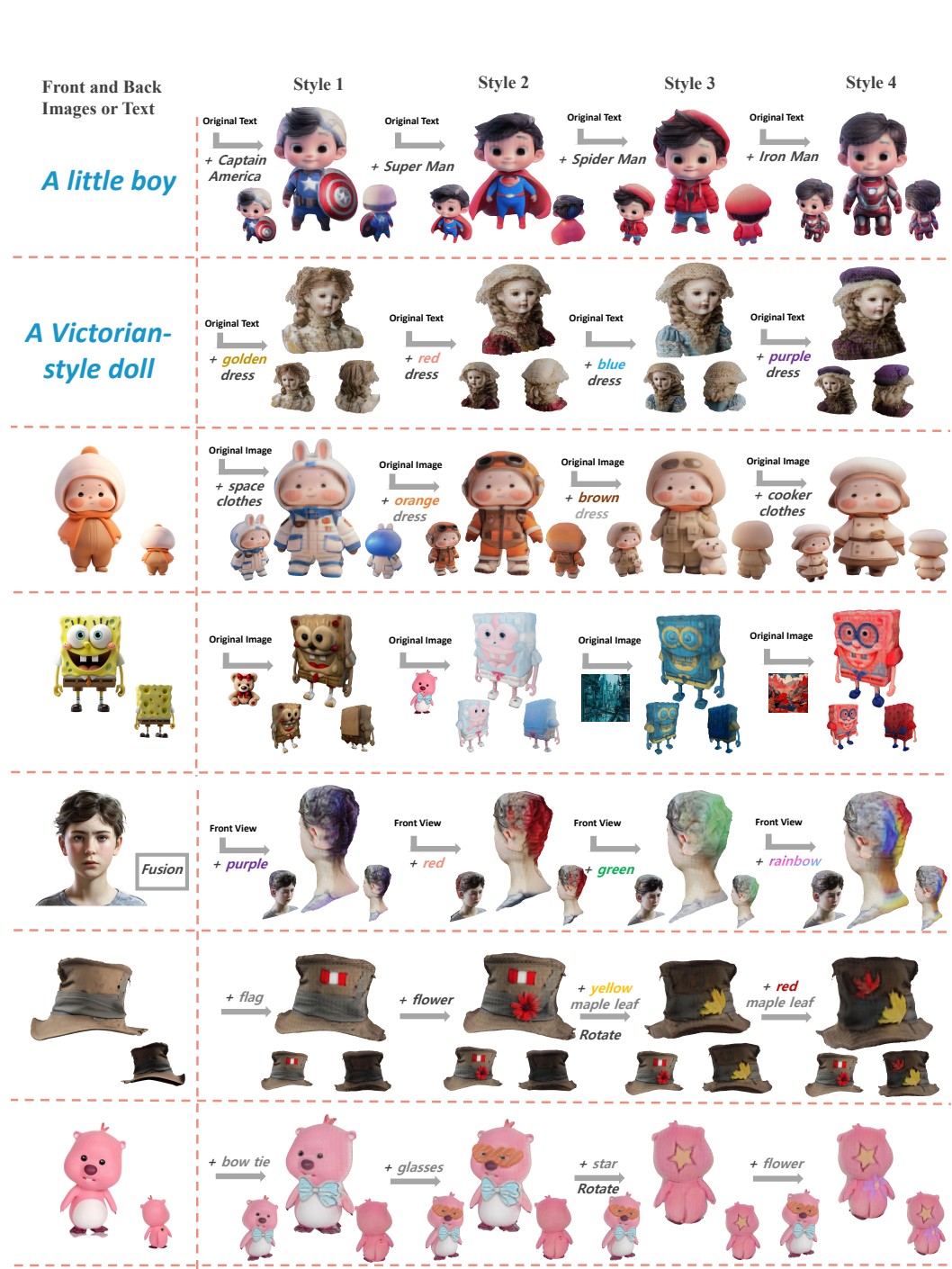

Figure 17: More Examples about Tailor3D.

