# OpenReview forum: "Tailor3D: Customized 3D Assets Editing and Generation with Dual-Side Images"
_ICLR.cc/2025/Conference — ICLR 2025 Conference Withdrawn Submission_

### Official Review · Reviewer_XARU · 2024-10-30

**Soundness:** 3
**Presentation:** 4
**Contribution:** 3
**Rating:** 6
**Confidence:** 2

**Summary:**

The paper introduced a practical system approach for 3D editing with the help of multi-view generations. The approach Tailor3D by generating the back view, then stitch the front and back view features for fine-tuning the reconstruction results. When testing on several datasets and benchmarks, the user studies showed the effectiveness of the approach, and it was quite competitive.

**Strengths:**

This is a nice system paper with quite some strengths
- The overall presentation of the approach is accompanied by good visualizations and diagrams. I find it useful understanding the whole paper
- The technique is interesting and it nicely integrated several technologies such as LRM, LoRA, and efficient attention mechanisms to edit and combine feature representations of front and back views.
- Overall, the result is competitive comparing to other approaches, it's nice to show user studies rather than automatically computed benchmark numbers in this paper

**Weaknesses:**

I think this is a nice system paper with good quality results. There are strong merits of this paper but I have a few questions
- I think the approach is heavily bounded by the quality of back-view generation algorithms. I wonder if authors could comment on whether there are empirical evidences or ablation studies on this topic. E.g: if the back-view generation algorithm is not performing well, would the front & back view fusion approach recover the issues?
- The approach is limited to using 2 views (front and back), would using more views increase the complexity of the formulations in this paper?
- In eq. 4, could authors help elaborating on the theory of the equation, or point to sections of the LRM paper that explains this equation? I think it might sound intuitive but I wonder if this is mathematically correct.

**Questions:**

My questions are listed in the weakness section

---

> ### Author Response · Authors · 2024-11-19
> **Reply to Reviewer XARU---Nov 19th**
>
> ## W-1: Does the proposed approach rely heavily on the quality of back-view generation, and are there empirical studies or ablations to address its impact on the fusion process?
>
> Your concern seems to be about conducting ablation studies on the back-view generation method. In our original paper, we used Zero-123, Stable-Zero123, and Zero123-XL, and our experiments showed that the 3D generation results were fairly similar across these methods. In fact, the resolution of the back-view images (224*224) generated through back-view generation methods is much lower compared to the front-view images(960*1200). Therefore, the performance of Tailor3D does not heavily rely on the quality of back-view generation.
>
> ## W-2: Would incorporating more views increase the complexity of the proposed approach?
> No, it won’t. We fuse the triplanes generated from different views, allowing us to directly utilize the pretrained LRM models. As a result, we only need to fine-tune the module responsible for fusing the triplanes, which does not increase the computational complexity.
>
> ## W-3: Equtation 4
> The meaning of Equation 4 is as follows: The initial triplane embedding is denoted as $f_{init}$. For both the front and back views, we set the camera extrinsics as $E_{t}$. The $'$ symbol represents the edited images. Here, $F_{f}^{'}$ refers to the features of the edited front-view image, which are used as input to generate the triplane $T_{f}$, the triplane derived from the front-view image. Similarly, $F_{b}^{'}$ refers to the features of the edited back-view image, which are used as input to generate the triplane $T_{b}^{f}$. However, $T_{b}^{f}$ needs to be rotated by 180 degrees to become $T_{b}$, which can then be fused with the triplane $T_{f}$.

---

> > ### Comment · Reviewer_XARU · 2024-11-26
> > **reply**
> >
> > I thank authors for providing the answers to my questions. I had a perhaps naïve question, what does "224224" and "9601200" mean in your first answer? also, what does "in our original paper" mean?

---

> > > ### Author Response · Authors · 2024-11-26
> > > **Reply to Reviewer XARU---Nov 26th**
> > >
> > > Dear reviewers
> > >
> > > That is the resolution which means 224 multiple 224 and 960 multiple 1200.
> > >
> > > Thanks

---

> > > > ### Comment · Reviewer_XARU · 2024-11-27
> > > > **reply**
> > > >
> > > > thanks -- what does "original paper" mean?

---

> > > > > ### Author Response · Authors · 2024-11-27
> > > > > **Reply to Reviewer XARU**
> > > > >
> > > > > Thanks for your reply!
> > > > >
> > > > > Sorry for the confusion. It means that in the  our main paper (Tailor3D), in the main pdf.
> > > > >
> > > > > Thanks!

---

> > > > > > ### Author Response · Authors · 2024-12-01
> > > > > > **Reply to Reviewer XARU**
> > > > > >
> > > > > > We have supplemented the experiments (compared to InstantMesh) and answered your questions. We are wondering if you have any further feedback or if there is a possibility of improving the score?

---

### Official Review · Reviewer_Sfh2 · 2024-10-31

**Soundness:** 2
**Presentation:** 2
**Contribution:** 2
**Rating:** 5
**Confidence:** 5

**Summary:**

This manuscript introduces a framework for generating customized 3D assets, which integrates 2D image editing methods with a Dual-side LRM. Initially, the front view is edited using 2D editing techniques. Subsequently, the back view is generated by a multi-view diffusion model and can also be edited using the same 2D method. Both views are then processed by the Lora Triplane Transformer to create the 3D object. The effectiveness of this method has been validated through extensive applications.

**Strengths:**

1. The proposed Dual-LRM and the editing of front and back views are technically sound. Editing these views without significant overlap effectively avoids inconsistent results. The proposed Dual-LRM can transform the two views into a 3D mesh.
2. The manuscript presents extensive editing results and an ablation study to validate each component within the framework.
3. The proposed LoRA Triplane Transformer and Fuse Double Side Feature effectively reduce the finetuning cost based on a pretrained LRM.

**Weaknesses:**

1. Grammar typo in Line 71.
2. Use \citep for citation.
3. The motivation in the contribution section is not clear. Dual-LRM is proposed to avoid inconsistency in multi-view image editing results. Why is there still inconsistency between the front and back views?
4. Too few comparison methods. InstantMesh[1] and CRM[2] also leverage multi-view images as input for 3D generation.
5. The technical novelty is a minor concern. The 2D image editing and multi-view diffusion model are directly used. The dual-sided LRM can also be directly replaced with InstantMesh. I recommend a comparison with InstantMesh by inputting edited two views into InstantMesh and Dual-side LRM.

[1]Xu J, Cheng W, Gao Y, et al. Instantmesh: Efficient 3d mesh generation from a single image with sparse-view large reconstruction models[J]. arXiv preprint arXiv:2404.07191, 2024.

[2] Wang Z, Wang Y, Chen Y, et al. Crm: Single image to 3d textured mesh with convolutional reconstruction model[J]. arXiv preprint arXiv:2403.05034, 2024.

**Questions:**

1. Compared to editing in a multi-view diffusion model, what are the advantages of editing in a single view? From my perspective, editing in a single view invariably incurs inconsistencies, though only front and back views are used, there is still an overlapped area for some objects.

2. Compared to training an LRM with only two views as input, what are the advantages of the proposed LoRA Triplane Transformer, except for memory cost? For example, how about the performance comparison? Moreover, since the LoRA Triplane Transformer is based on a pretrained LRM, how does the performance of directly training a Dual-side LRM from scratch compare?

---

> ### Author Response · Authors · 2024-11-19
> **Reply to Reviewer Sfh2---Nov 19th**
>
> Thank you very much for your response. Let me address some of your questions below.
>
> ## W-1&2: Some typo about the paper.
> Thank you so much for reading so thoroughly; we will make sure to address and correct this in future versions.
>
> ## W-3: Why does inconsistency still exist between the front and back views if Dual-LRM aims to address multi-view editing inconsistencies?
> Our explanation might not have been clear enough—multi-view inconsistency actually has two distinct aspects.
>
> - The first aspect is multi-view inconsistencies during the reconstruction process. If the front and back images are inconsistent (e.g., the cameras are misaligned), Dual-LRM is capable of handling this issue.
>
> - The second aspect is multi-view inconsistencies during editing. Editing multiple views simultaneously is challenging—for instance, if I want to change someone's outfit, it’s quite difficult to ensure that all views are updated consistently to reflect the change. By using only the front and back views, we can largely avoid this issue.
>
>
> ## W-4: The paper lacks sufficient comparisons, as methods like InstantMesh and CRM also use multi-view images for 3D generation.
> First, our method is not aimed at enhancing the quality of 3D object generation but rather at providing an editing approach. Additionally, both CRM and InstantMesh use Zero-123++ to generate six views from a single image before reconstruction, so the setting is somewhat different.
>
>
> ## W-5：The technical novelty is limited, and a comparison with InstantMesh using edited two-view inputs is recommended.
> Our initial intention was to avoid re-pretraining. We aimed to achieve results by simply fine-tuning the model parameters of LRM. In contrast, methods like Instant3D and InstantMesh require re-pretraining.
>
> Additionally, InstantMesh requires six fixed-view inputs, which do not include the front and back views. As a result, this approach cannot be directly applied to InstantMesh. However, as you mentioned, multi-view image editing is indeed a very critical direction for future work.

---

> > ### Comment · Reviewer_Sfh2 · 2024-11-21
> >
> > Since you claim a contribution of the Dual-sided LRM in improving overall reconstruction quality, I assume your primary goal is to enhance reconstruction quality. If that is the case, InstantMesh also supports two views as input. Therefore, I believe you should include a comparison with InstantMesh to evaluate performance under similar conditions.
> >
> > Moreover, if InstantMesh with two edited views already achieves good 3D reconstruction, I question whether the Dual-sided LRM is necessary. Could you clarify how it provides additional benefits compared to InstantMesh in this scenario?

---

> > > ### Author Response · Authors · 2024-11-21
> > > **Reply to Reviewer Sfh2---Nov 21th**
> > >
> > > Thanks, I will demonstrate more examples of what our method can achieve that InstantMesh cannot.

---

> ### Author Response · Authors · 2024-11-19
> **Reply to Reviewer Sfh2---Nov 19th**
>
> ## Q-1: What are the advantages of single-view editing over multi-view diffusion models, given the potential for inconsistencies even with front and back views?
> As mentioned in the paper, methods using multi-view diffusion primarily optimize for each individual object or scene rather than serving as a general framework. Similar to NeRF, these approaches require optimization for each object to be edited. In contrast, the LRM-based method is a general framework that uses a feed-forward approach. Additionally, as shown in Figure 8, we provide a comparison with MVEdit (multi-view diffusion). Moreover, multi-view diffusion currently only supports text-driven editing and cannot modify specific parts of an object.
>
>
> ## Q-2: What are the advantages of the proposed LoRA Triplane Transformer over training an LRM with two-view inputs, beyond memory cost, and how does its performance compare to directly training a Dual-side LRM from scratch?
> The main purpose of using LoRA here is to reduce memory usage. The original pretraining structure was overly redundant and required 64 A100 GPUs for pretraining. After adopting the LoRA structure, our model can be fine-tuned on a 4090 GPU. I forgot to mention this in the paper because we used A100 GPUs at the time to speed up the experiments.

---

> ### Author Response · Authors · 2024-11-26
> **Reply to Reviewer Sfh2: Question about InstantMesh ---Nov 26th**
>
> We greatly appreciate your response. Regarding your question, we have conducted additional experiments, and the results are presented in the **PDF of the supplementary materials**. In **Figure 1**, we show the back-view editing process of InstantMesh as you mentioned, along with our own editing process. The results demonstrate that InstantMesh has very high requirements for viewpoint consistency. After editing, when the images are reconstructed using InstantMesh, they fail to produce correct results. For example, the sofa shows holes and missing parts. In contrast, Tailor3D successfully avoids these issues and produces a more reliable reconstruction.
>
> In **Figure 2**, we present more examples. It is important to note that these examples are all from the original paper and are not cherry-picked. In addition to the failures caused by viewpoint inconsistencies, InstantMesh can even alter the original shape of the object. For example, in the case of Captain America, the original input image shows that the figure's legs are short and thick, but after processing with InstantMesh, the legs become long and thin. Tailor3D, on the other hand, better preserves the features from the original input image.

---

### Official Review · Reviewer_5Wus · 2024-11-01

**Soundness:** 2
**Presentation:** 2
**Contribution:** 1
**Rating:** 3
**Confidence:** 3

**Summary:**

This paper proposes Tailor3D, a method for fast 3D generation and editing.
The core component of Tailor3D is a dual-side image LRM, which takes a dual-side image pair (front and back) as input, and output a 3D representation of the object (in Triplane).
Compared to original LRM, this dual-side LRM takes two images, instead of a single image or four images.
The motivation of using dual-side image is that, using two images, especially front and back views can provide convinience for users to edit while preventing critical inconsistencies from more views (like 4 views).
This paper show extensive experiments about dual-side image LRM, including generation, editing, and very dense ablations.

**Strengths:**

1.This paper introducing a very interesting sub-problem: given a pretrained LRM, how could we merge the the output triplane of dual-side image into a single triplane. This paper proposes a sound method to do so and shows the effectiveness in the experiments.
2. The ablation study in this paper is quite comprehensive, including the way of merging two triplanes, the rank of LORA, extrinsic parameters, etc.
Also, this paper provides the failure cases and analysis, which is very useful for understanding the model.
3. This paper is easy to read and follow.

**Weaknesses:**

1. My biggest concern about this paper is that many considerations lack clear motivation, which makes me wonder whether some problems addressed are real or artificial.

(1) This paper emphasizes reducing the memory usage of LRM and therefore proposes using LoRA for a pretrained LRM. However, I don't see a clear reason why memory usage would be problematic here. Given that there are even four-view versions of LRM [DMV3D: DENOISING MULTI-VIEW DIFFUSION USING 3D LARGE RECONSTRUCTION MODEL] and this paper uses an A100 GPU, I don't understand why memory usage is a critical concern and whether LoRA is a necessary component for the model.

(2) Similarly, this paper proposes a novel method to merge two triplanes using cross-attention. I like this solution. However, I don't see a clear justification for this approach. In particular, why not simply take the features from both images and fine-tune the LRM model to generate a single triplane? This pipeline seems more natural and easier to implement. I may be missing some important considerations here.

(3) The dual-side image setting also seems somewhat unusual. While I agree that using two images can help eliminate inconsistencies across multiple views, it also constrains general editing capabilities. For example, if users want to edit the side of an object (between the front and back views), the current setting may be inconvenient. This raises the question: is the current setting really necessary? Could we instead use four views, edit in one of them, and fine-tune LRM to propagate these changes into triplanes?

It would be greatly appreciated if the authors could address these questions in their rebuttal. More importantly, the paper would benefit from clearer motivation and more detailed elaboration of these considerations to help readers better understand both the problems and proposed solutions.

2. Table 1 is confusing to me. What's the setting of single-image generation here? How could the proposed method achieve this task as the trained LRM takes dual-side images as input? If just providing single image, why this model outperforms the others so much as it should be similar to the original LRM with single image input?

3. I understand that training LRM may take a long time and many resources. But I still think it might be worthwhile to do some higher-level ablations to support some design considerations, mentioned in the first point of weakness. For example, could we train a LRM taking the image embedding from two images and directly outputs the triplane? Would that be better or worse compared to the current one? I think this paper does a great job on ablating the components in the current design, but some higher-level ablations would be really needed to support some decisions.

**Questions:**

I have some questions listed in the weakness part. I would be very happy to change my rating given the answers of these questions.

---

> ### Author Response · Authors · 2024-11-18
> **Reply to Reviewer 5Wus---Nov 18th**
>
> Thank you so much for reading our paper so thoroughly. Your insights and reflections on the work even surpass my own as the author! These are incredibly constructive comments!
>
> ## W-1(1): Why is memory usage a critical concern, and is LoRA truly necessary for the model?
>
> The purpose of using LoRA is to **reduce memory usage**. Tailor3D can be fine-tuned on an RTX-4090. Tailor3D is essentially a fine-tuned version of the LRM framework. However, to accelerate the experiments later on, we used an A100.
>
>
>
> ## W-1(2): Why not directly fine-tune the LRM model using features from both images to generate a single triplane, as it seems more natural and straightforward?
>
> Concatenate the image features of the front and back views, or even more views (such as the four views in DMV3D), and then use a transformer to generate a triplane. This is indeed the approach adopted by most papers currently.
>
>
> While working on this project, the third-party reproduction code for LRM was the only available open-source implementation, as none of the other methods had been made publicly accessible. We also aimed to extend the LRM from single-view to multi-view efficiently. However, as illustrated in Figure 10, a fundamental difference exists between LRM and Instant3D (MV-LRM) in how they handle camera parameters.
>
> - LRM injects the camera parameters into the image-to-3D triplane process.
> - Instant3D integrates the camera parameters into the image backbone.
>
> These structural differences make it infeasible to directly fine-tune LRM to produce Instant3D’s outputs.
>
> So we considered an alternative approach: generating two triplanes from two images and then combining these triplanes at the 3D level to integrate the information. **This design allows us to leverage the pretrained LRM model parameters and achieve two-view results with minimal fine-tuning.** Additionally, no prior work has explored fusing multi-view information directly at the triplane level, which motivated us to attempt this approach.
>
> ## W-1(3): Is the dual-side image setting truly necessary, or could using four views with fine-tuning for edits be a more flexible solution?
>
> You're absolutely right—using two views does seem a bit unusual. The reasons we chose this paradigm are as follows.
>
> - **Initial exploration:** Tailor3D is an initial exploration, and focusing on the front and back views is likely the simplest scenario.
>
> - **Sufficiency:** Editing with just the front and back views is sufficient to meet the editing needs of most objects. Achieving highly accurate editing across four views is very challenging due to the lack of multi-view consistent image editing approaches.
>
> - **Experimentation:** We attempted generating four triplanes from four images (front, back, left, and right) and combining them into a single triplane through rotation, but this approach proved unsuccessful.
>
> Currently, rotating triplanes only works for a 180-degree rotation (i.e., between the front and back triplanes). **Rotating the left and right triplanes by 90 degrees is not feasible**, as the features become misaligned after rotation.
>
> ## W-2: How does the proposed method achieve single-image generation and why does it outperform others despite relying on dual-side inputs?
>
> For Tailor3D, the front-view image is used as input, while the back-view image is generated using multi-view image generation algorithms like Zero-1-to-3[1]. These front and back views are then fed into the model. Similarly, other methods in the table, such as TriplaneGaussian, Wonder3D, and LGM, follow a similar approach: a single front-view image is used to generate 4-6 additional views, and these generated images are then used for reconstruction. While this approach may seem somewhat unusual, it is indeed categorized as a single-image method.
>
> ## W-3: Could higher-level ablations, such as training LRM to directly output a triplane from two image embeddings, better support the design decisions in the paper?
>
> You make an excellent point. Our motivation at the time was to fine-tune LRM to achieve our current results. If we were to merge 2D features, it would require large-scale pretraining from scratch **as mentioned in W-1(2)**, which would be extremely costly. Do you think this experiment is necessary? If you feel it is, I can consider adding this ablation experiment.
>
>
>
>
>
> [1] Zero-1-to-3: Zero-shot One Image to 3D Object.

---

> ### Comment · Reviewer_DNWZ · 2024-11-18
> **reply.**
>
> I just want to say that, regarding the response of why dual-side is necessary, I think using "this is an initial exploration" is NOT really convincing and weird response...
> If a setting is not a valid and reasonable setting, then I think this setting should be discarded during initial exploration and work on a more reasonable setting...
> Overall, I am not saying in this case the two view setting is really a bad idea, just try to point out that the response makes me even unsure whether the setting is reasonable or not...

---

> > ### Author Response · Authors · 2024-11-19
> > **Reply to Reviewer 5Wus---Nov 19th**
> >
> > Thank you very much for your prompt response. I realize that replying with too much information at once might make things a bit unclear, so let me explain the reasons step by step.
> >
> > First, one of the key reasons is that this setting allows us to leverage the pretrained results of LRM without the need for re-pretraining, as shown in W-1(2). With just a single GPU, we can fine-tune the model to achieve dual-view results.
> >
> > In contrast, if we were to concatenate multiple image features and generate a single triplane (like Instant3D), it would require complete re-pretraining. This pretraining process demands 64 A100 GPUs running for a week to complete.
> >
> >
> > This is one of the key reasons why we chose this setting. Do you have any questions about this point, or is there anything unclear in my explanation?

---

> > ### Author Response · Authors · 2024-11-19
> > **Reply to Reviewer 5Wus---Nov 19th**
> >
> > **Your Question: Could we instead use four views, edit in one of them, and fine-tune LRM to propagate these changes into triplanes?**
> >
> > First, LRM can only accept a single image and cannot handle four images. Are you referring to Instant3D (4-view LRM)?
> >
> > **Let me provide an example that I think can serve as a basis for our discussion.**
> > Suppose we have an image of a Mario figurine, and we want to generate and edit it in the form of four views as you suggested.
> >
> > We can generate its four views based on the front view. Now, if you wish to edit any of these views, for instance, editing the right view to make Mario hold a ball, while the front, back, and left views remain unchanged, this would immediately cause inconsistencies. This would result in unacceptable outputs during both fine-tuning and inference.
> >
> > I would also like to clarify that the inconsistency addressed by Tailor3D refers to the inconsistency in camera extrinsics. For example, suppose we have front and back-view images of the Mario, but the front and back images are not captured from strictly aligned front and back perspectives. Tailor3D can accept images with non-strictly aligned camera extrinsics and still produce a reasonable 3D mesh.
> >
> > However, the entire LRM series cannot handle inconsistencies in content across different views. For example, if Mario is holding a ball only in the right view while the other views show him without the ball, inputting such data into the model would fail to produce a satisfactory 3D mesh. This is also why we chose the dual-view setting. First, the front and back views of most objects contain the core information. Additionally, because they have minimal overlap, they can be edited independently. The edited front and back images can then be processed using Tailor3D to generate the 3D mesh result.
> >
> > I hope I was able to address your concerns and look forward to further discussions with you. I can create a figure later to help explain this issue.

---

> > > ### Comment · Reviewer_5Wus · 2024-11-23
> > > **Reply to the rebuttal**
> > >
> > > Thanks for the rebuttal and it makes lots of things more clear.
> > >
> > > As for the motivation of finetunning, fusing existing two LRMs instead of training some multiview variants of LRM, I personally think it is fine to me. It is not satisfying as I think the current solution is very temporary and will be replaced by some more general solutions very soon, but I also understand doing research is inevitable to be limited by current available code and resources. Is it Oaky to understand your project as "targeting to finetune an existing LRM to achieve the front-back view settings with minimal efforts"? If so, could you please provide some numbers to show the advantages of this finetunning? Like finetunning time? Memory requirements (like work on 4090) or some other benefits?
> > >
> > > For the settings of front-back view, I agree with Reviewer DNWZ that saying something is an initial exploration is not convincing. But if you can show this setting has an advantage for your finetunning, I might be ok with it.

---

> > > > ### Author Response · Authors · 2024-11-23
> > > > **Reply to Reviewer 5Wus---Nov 24th**
> > > >
> > > > Dear Reviewer 5Wus
> > > >
> > > > Thanks for your reply, we are answering the question of Sfh2 now. In the response, we compare with the SOTA method Instantmesh. We will show more failure cases about it and explain why we need this setting.
> > > >
> > > > Also, the Instantmesh has dual-sided setting as well, we will compare to it. We will respond to all of you with qualitive examples tomorrow.
> > > >
> > > > Thanks a lot!

---

> > > > > ### Author Response · Authors · 2024-11-26
> > > > > **Reply to Reviewer 5Wus---Nov 26th**
> > > > >
> > > > > We greatly appreciate your reply once again.
> > > > >
> > > > > First, the motivation behind our initial exploration is to avoid over-claiming. Personally, I believe many papers today exaggerate their advantages, and we aim to present our work more objectively. It is important to note that all the qualitative experimental results in this paper are not cherry-picked. We hope to make a substantial contribution to the 3D generation community.
> > > > >
> > > > > ## **1. Memory Usage and Training Time**
> > > > > Regarding memory and training time, our model is divided into three configurations: small, base, and large. The parameters for each model are as follows:
> > > > >
> > > > > | Model            | Pretained Model        | Layers | Feat. Dim | Trip. Dim. | In. Res. | Image Encoder     |
> > > > > |------------------|------------------------|--------|-----------|------------|----------|-------------------|
> > > > > | tailor3d-small-1.0 | openlrm-mix-small-1.1  | 12     | 512       | 32         | 224      | dinov2_vits14_reg |
> > > > > | tailor3d-base-1.0  | openlrm-mix-base-1.1   | 12     | 768       | 48         | 336      | dinov2_vitb14_reg |
> > > > > | tailor3d-large-1.0 | openlrm-mix-large-1.1  | 16     | 1024      | 80         | 448      | dinov2_vitb14_reg |
> > > > >
> > > > > The parameters and the size of our finetuned model are as follows:
> > > > >
> > > > > | Model            | Model Size | Model Size (with pretrained model) |
> > > > > |------------------|------------|-------------------------------------|
> > > > > | tailor3d-small-1.0 | 17.4 MB   | 436 MB                              |
> > > > > | tailor3d-base-1.0  | 26.8 MB   | 1.0 GB                              |
> > > > > | tailor3d-large-1.0 | 45 MB     | 1.8 GB                              |
> > > > >
> > > > > As can be seen, the size of our finetuned model only accounts for **3% of the total parameters**. For the base configuration, it takes approximately **36 hours to train on a single 4090 GPU** using Gobajverse-LVIS. Therefore, our model is suitable for finetuning in a laboratory setting.
> > > > >
> > > > > ## **2. Why Dual-views Setting is Important?**
> > > > > We also need to elaborate on the benefits and motivation for using front-back views. **In the latest supplementary material, we have included experiments comparing our method with InstantMesh.** This also addresses Reviewer Sfh2's comment, as InstantMesh is currently the state-of-the-art open-source work for multi-view 3D generation.
> > > > >
> > > > > The pipeline of editing with InstantMesh is shown in **Figure 1.** First, InstantMesh starts with a single image and uses Zero-123++ to generate six fixed viewpoints. Then, editing is performed on the two back viewpoints, and finally, the edited six viewpoints are used for reconstruction. In contrast, our Tailor3D uses a dual-view setting. Here, we input a front view image, then use Zero-123 to generate the back view of the object. After editing the back view, we use Tailor3D for reconstruction.
> > > > >
> > > > > Comparing the results with the sofa, we found that after editing the back view, InstantMesh exhibits inconsistencies in the back view images. This directly leads to failures in 3D reconstruction, such as holes appearing in the sofa. *In contrast, our method ensures better consistency across viewpoints, resulting in satisfactory results.*
> > > > >
> > > > > We show more examples in **Figure 2**, such as Captain America and our small bird. It should be noted that none of our examples are cherry-picked; all the examples are from the original paper. It is clear that using the InstantMesh framework, which has high consistency requirements for viewpoints, leads to inconsistencies in the edited images, and the reconstructed 3D objects fail. In our example, the front and back views of the bird have different styles. However, InstantMesh not only fails to perform style fusion, but also introduces defects and errors in the shape of the bird itself.

---

> > > > > > ### Author Response · Authors · 2024-12-01
> > > > > > **Reply to Reviewer 5Wus---Dec 01st**
> > > > > >
> > > > > > We have answered your question and supplemented it with experiments. We have provided the model size, training time, etc. Additionally, we have included results from InstantMesh for comparison. We believe a score of 3 for rejection is indeed too low for this paper.

---

> ### Author Response · Authors · 2024-11-19
> **Reply to Reviewer 5Wus---Nov 19th**
>
> Since my initial response needed to address all of your questions comprehensively, some details may have been overlooked. Here, we will focus on addressing the first question in detail.
>
> You're absolutely right, using two views does seem a bit unusual. By saying it is a intial exploration, we does not mean that the proposed scheme is built on a non-realistic foundation. Instead, we think it is a simple, but indeed reasonable (enough for reconstruction using double-side information) and efficient (requires less information compared with other multi-view schemes).
>
> **It is impossible to fuse image features and generate a single triplane through fine-tuning alone; pretraining is required.**
>
> The structures of LRM and DMV3D (which can also be considered a type of MV-LRM) are significantly different. Please note that in LRM, the camera parameters are input into the triplane transformer in a modulated form, whereas DMV3D and Instant3D do not use this approach. This difference makes the two frameworks independently pretrained frameworks. Therefore, as you suggested, directly concatenating image features to generate a single triplane would require re-pretraining, which would take at least 64 A100 GPUs for a week on Objaverse. In contrast, Tailor3D can achieve results similar to DMV3D by simply fine-tuning on top of LRM. (Moreover, neither DMV3D nor Instant3D has been open-sourced.)
>
> Tailor3D is a fast and efficient module that extends LRM or any single-view reconstruction framework into a dual-view editing framework. Our core contribution lies in proposing this pipeline, which can be applied to extend any single-view reconstruction into a dual-view editing framework. Dual-view editing is already sufficient to meet the requirements of most 3D editing tasks.

---

### Official Review · Reviewer_DNWZ · 2024-11-02

**Soundness:** 3
**Presentation:** 2
**Contribution:** 2
**Rating:** 5
**Confidence:** 3

**Summary:**

This paper presents a method that allows users to customize 3D assets using 2D edits through a dual-side triplane-NeRF generation model that takes both front-view and back-view as input. And they perform thorough studies to evaluate the proposed method.

**Strengths:**

* The input on both front-view and back-view seem reasonable and achieve good quality results.
* The evaluation seems thorough.

**Weaknesses:**

* The method is a combination of existing methods and trained on a new data format that involves both front and back view. For me, it is unclear what is the major technical contribution.
* The quality of the generated 3D results are poor... I think this is due to the backend (LRM), but I still think it is worth pointing out that so many people nowaday relying on these poor 3D generative model for different purpose, but the reality might be that the quality of the 3D content generated by these models are just not good enough for real-world applications.

**Questions:**

* Following the point mentioned in the weakness seciton, I wonder whether it is possible to extend or generalize the proposed method to the future 3D generative model? Since I think if not, then for real-world scenario, the proposed method is not useful at all because of limited 3D content quality bounded by the backbone model.
*

---

> ### Author Response · Authors · 2024-11-19
> **Reply to Reviewer DNWZ---Nov 19th**
>
> ## W-1: What is the key technical contribution, given that the method combines existing approaches and uses a new front-and-back view data format?
> As you mentioned, our core idea focuses on a dual-view image editing paradigm and the method for fusing triplanes obtained from different perspectives, rather than pursuing the highest generation quality.
>
> ## W-2 & W-3: The generated 3D results are poor, highlighting concerns about the reliance on low-quality 3D generative models for real-world applications.
>
> Tailor3D is not about achieving the highest generation quality but is designed to be compatible with any 3D generation method based on triplanes. Therefore, it can be extended to most 3D object generation frameworks. Additionally, this structure can be fine-tuned on a personal computer without the need for large-scale pretraining. For these reasons, we believe Tailor3D holds practical value for real-world editing applications.

---

> ### Author Response · Authors · 2024-11-28
> **Reply to Reviewer DNWZ---Nov 26th**
>
> May I ask why you changed it from 6 to 5? Did my response not meet your expectations?

---

### Comment · Area_Chair_GM6Q · 2024-11-25
**Please read the rebuttal and reply**

Dear Reviewers,

Thanks again for serving for ICLR, the discussion period between authors and reviewers is approaching (November 27 at 11:59pm AoE), please read the rebuttal and ask questions if you have any. Your timely response is important and highly appreciated.

Thanks,

AC

---

### Note · Authors · 2025-02-06

I have read and agree with the venue's withdrawal policy on behalf of myself and my co-authors.

---

### Meta-Review · Area_Chair_GM6Q · 2024-12-23

**Metareview:**

This paper proposes a method for rapid 3D object editing. The main idea is to leverage 2D editing models and 3D reconstruction models. Specifically, given a front view image as input, it first edits the image with editing methods, then synthesizes back-view image using MV diffusion models and edit it similarly, finally, the paper proposes a dual-side LRM to reconstruct the edited 3D objects. To extend the LRM to include back-view as input, the paper adopts LoRA as well as a fusing mechanism.

This paper is well-written and easy to follow, the task is interesting and the authors have done comprehensive experiments to show the effectiveness of the proposed method.

For weakness, the main concerns raised by reviewers are
- the dependency on existing works and the limitation of the technical contribution of the paper. (Reviewer DNWZ, Sfh2, XARU)
- the motivation of the current setting (e.g., if dual-side setting is reasonable?) (Reviewer 5Wus)
- missing comparisons (Reviewer Sfh2)

The authors have actively answered these questions with new experiments (see the Reviewer Discussion below), unfortunately reviewers are not fully convinced and three out of four reviewers lean to reject this paper.

After reading the paper, reviews and rebuttals, AC agrees with the reviewers on the following points: the paper is dependent on existing modules and the motivation is not fully clear. Specifically, the editing part is done through off-the-shelf editing methods, making the contributions focus on the dual-side reconstruction method. Based on this, AC agrees with Reviewer 5Wus that the setting is questionable (or at least the claim of the paper may focus more on reconstruction instead of editing) and Reviewer Sfh2 that the reconstruction quality does matters and should be thoroughly compared with existing methods.

To conclude, the paper is recommended for rejection for this time and the authors are encouraged to improve it based on all feedbacks.

**Additional Comments On Reviewer Discussion:**

The main points raised by reviewers include the dependency on existing works and the limitation of the technical contribution of the paper. (Reviewer DNWZ, Sfh2, XARU) and the motivation of the current setting (Reviewer 5Wus). Though during rebuttal, authors have discussed their opinions on these points and add more experimental results, as discussed above, these points are not fully addressed. After rebuttal, three out of four reviewers lean to reject the paper.

---

### Decision · Program_Chairs · 2025-01-22

Reject